# BLENDERFUSION: 3D-GROUNDED VISUAL EDITING AND GENERATIVE COMPOSITING

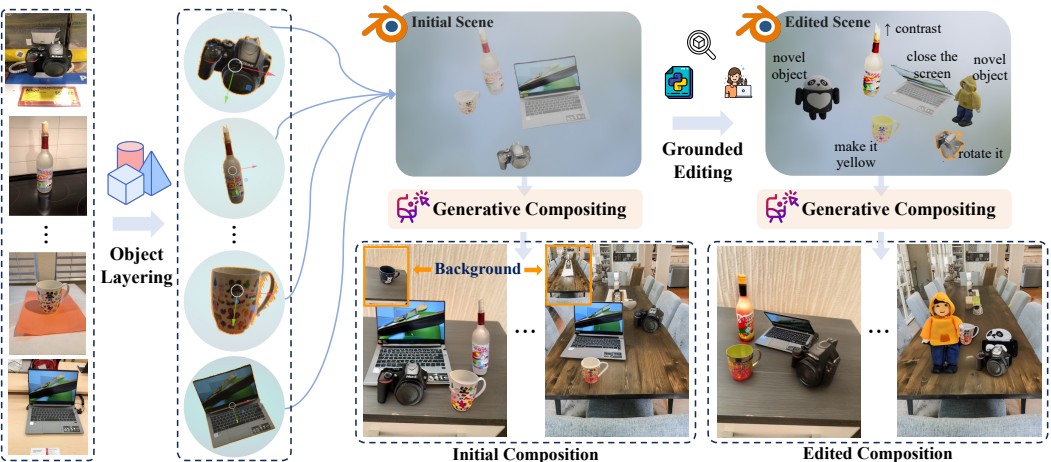

Figure 1: BlenderFusion integrates the 3D-grounded editing capabilities of graphics software into the strong synthesis abilities of diffusion models. It enables precise object and camera control, inherits Blender's rich editing functionalities (e.g., attribute change, part deformation), and generalizes to highly fine-grained multi-object editing and scene composition tasks (Figure 6).

## ABSTRACT

We present BlenderFusion, a generative visual compositing framework that recomposes objects, camera, and background to synthesize new scenes. It follows a layering-editing-compositing pipeline that (i) segments and converts visual inputs into editable 3D entities (layering), (ii) edits them in Blender with 3D-grounded control (editing), and (iii) fuses them into a coherent scene using a generative compositor (compositing). The generative compositor extends a pre-trained diffusion model to process both the original (source) and edited (target) scenes in parallel, and is fine-tuned on video frames with two important training strategies: (i) source masking, enabling flexible modifications like background replacement; (ii) simulated object jittering, facilitating disentangled control over the objects and camera. Extensive experiments on synthetic and real-world datasets show that BlenderFusion significantly outperforms prior methods in precise 3D-aware control and complex compositional scene editing. The framework also generalizes to unseen data and fine-grained editing operations beyond the training distribution.

## 1 INTRODUCTION

*Visual compositing* is the process of constructing a scene by extracting objects from multiple images, manipulating their appearance or spatial configuration, inserting them into a new background, and adjusting the camera to produce a cohesive image or video. This enables novel visual narratives with high flexibility. While recent generative AI techniques excel at photorealistic text-to-image synthesis (Karras et al., 2020; Rombach et al., 2022; Saharia et al., 2022; Ramesh et al., 2022; Baldridge et al., 2024), they often fall short in complex compositing scenarios that demand precise, 3D-aware control over multiple scene elements, such as repositioning objects, modifying geometry and appearance, and adjusting viewpoint consistently.

Table 1: The capability comparisons between BlenderFusion and existing representative 3D-aware editing methods: Object 3DIT (Michel et al., 2024), Neural Assets (Wu et al., 2024), and Image Sculpting (Yenphraphai et al., 2024)

| Method | Visual Elements | | | Object Control | | | | Control Interface |
|---|---|---|---|---|---|---|---|---|
| | Obj | Cam | BG | Multi-Obj | Novel-Obj | Attribute Change | Non-rigid Trans. | |
| Object 3DIT | ✓ | ✗ | ✗ | ✗ | ✗ | ✗ | ✗ | Text |
| Neural Assets | ✓ | ✓ | ✓ | ✓ | ✗ | ✗ | ✗ | Obj tokens |
| Image Sculpting | ✓ | ✗ | ✗ | ✗ | ✓ | ✓ | ✓ | Blender |
| BlenderFusion (Ours) | ✓ | ✓ | ✓ | ✓ | ✓ | ✓ | ✓ | Blender |

To better characterize visual compositing tasks, we consider two key aspects: (i) the editable visual elements—objects, camera, and background—and (ii) the granularity of object-level control, including multi-object editing, novel object insertion, attribute modification, and non-rigid transformations. Some recent approaches blend objects from multiple input images while preserving identity and following coarse geometric layouts (Hu et al., 2024a; Song et al., 2022), but their control remains implicit, lacking 3D awareness and disentanglement. A more explicit line of work incorporates 3D-aware mechanisms to enhance editing fidelity and controllability.

Object 3DIT (Michel et al., 2024) enables text-driven 3D-aware edits with synthetic training data but is limited to single-object rigid transformations. Neural Assets (Wu et al., 2024) disentangles object and background tokens for multi-object composition and manipulation, yet lacks fine-grained control. Image Sculpting (Yenphraphai et al., 2024) leverages Blender for accurate 3D edits but requires per-scene optimization and is limited to editing a single object from a single image. Table 1 summarizes these representative methods across the two aspects above, highlighting that none achieves full-scene visual compositing with fine-grained, disentangled control over all core elements.

To address these limitations, we propose BlenderFusion, a unified framework that mirrors the traditional visual compositing process through three fundamental steps (Figure 1). (1) *Layering*: off-the-shelf vision foundation models delineate foreground objects and lift them into 3D-editable entities. (2) *Editing*: native Blender operations enable fine-grained, 3D-grounded modifications of object geometry, appearance, camera viewpoint, etc. (3) *Compositing*: a diffusion-based visual compositor incorporates editing intents and fuses the Blender renders with a background to produce a coherent image of the edited scene. By decoupling 3D control from image generation, BlenderFusion combines the strengths of graphics-based editing and generative synthesis, enabling flexible, disentangled, and 3D-aware manipulation of objects, camera, and background.

Concretely, our diffusion compositor (Figure 2) operates on two parallel input streams: a source stream containing the original scene and a target stream reflecting the edited scene. To train the compositor effectively, we leverage object-centric videos and introduce two training strategies: (1) *Source masking*: To handle significant contextual changes such as object insertion, replacement, or removal, we mask the modified regions in the source stream, preventing them from interfering with target composition. At test time, this allows flexible masking to expose only the valid context. (2) *Simulated object jittering*: Since training videos are often dominated by camera motion, we introduce a reconstruction training setup that jitters object positions between source and target renders while keeping the camera fixed, thereby enriching supervision for disentangled object control.

We evaluate our method on three datasets–MOVi-E (Greff et al., 2022), Objectron (Ahmadyan et al., 2021), and Waymo (Sun et al., 2020). Although these datasets only exhibit basic object and camera motions, our method generalizes to diverse editing scenarios, enabling precise disentanglement of visual elements, complex compositional editing, and generalization beyond training data and editing operations (Figure 1, Figure 6, and Figure 8). Our contributions are summarized as follows:

**1)** A generative visual compositing framework, *BlenderFusion*, that integrates the best of both worlds: the accurate and 3D-grounded editing functionalities of a graphics engine (e.g., Blender), and the strong synthesis capability of state-of-the-art generative AI (e.g., diffusion models).

**2)** A novel dual-stream diffusion compositor that synthesizes the target image from 3D-grounded Blender renders of the source and edited scenes, accompanied by two carefully designed training strategies that facilitate the learning of disentangled and flexible 3D-aware control.

**3)** State-of-the-art results on disentangled control over objects and camera, fine-grained multi-image and multi-object compositing, and generalization to unseen objects and editing tasks. Please check out our supplementary material for *videos* and *interactive results* (See §A for details).

## 2    RELATED WORK

**Visual Generation and Control.** Modern diffusion-based generative models (Sohl-Dickstein et al., 2015; Ho et al., 2020; Song et al., 2020) have achieved high-fidelity content generation. The focus then shifted to controllability, with natural language emerging as the primary interface (Ramesh et al., 2022; Rombach et al., 2022; Saharia et al., 2022). However, text-based control is inherently limited. To address this, specialized methods have been developed for more granular control over aspects such as geometry (Zhang et al., 2023b), subjects (Ruiz et al., 2023; Chen et al., 2022; 2023), and aesthetics (Sohn et al., 2023). Recent efforts have focused on unifying these disparate controls into a single, powerful model (Hu et al., 2024a). However, the reliance on text control interface persists, along with its inherent ambiguity and difficulty in articulation. Our work addresses this by augmenting the generative model with a graphics engine, which leverages the generation strength of diffusion models while gaining the precise control of a graphics engine.

**Visual Compositing.** Visual compositing—the process of assembling various visual elements into a seamless, photorealistic image—is a crucial yet challenging task in visual generation. Early diffusion-based methods focused on simple, single-object compositions guided by layouts (Lu et al., 2023; Song et al., 2024; Chen et al., 2024; Zhang et al., 2023a; Yuan et al., 2023). While recent works handle multiple objects (Tarrés et al., 2025) or even manipulate semantic attributes by mixing concepts from different images (Garibi et al., 2025), these approaches remain confined to the 2D space. A comprehensive suite for complex, multi-object, and multi-image composition with precise geometric control remains an underexplored problem, which is the focus of this paper.

**3D-aware Control.** 3D-aware control that respects the physics and geometry is a fundamental challenge in visual generation. Existing methods are primarily limited to single-object manipulation and often entangle visual elements, making complex editing difficult. GeoDiffuser (Sajnani et al., 2025) and Diffusion Handle (Pandey et al., 2024) use depth priors to transform model activations, while Magic Fixup (Alzayer et al., 2024) implicitly learns physics from video frames. More recently, 3D-Fixup (Cheng et al., 2025) explicitly learn this transformation through 3D edits. A key limitation, however, is that these methods are all designed for single-object control. As noted in Table 1, NeuralAsset (Wu et al., 2024) is a notable effort that addresses multi-object control, including background and camera. Our framework provides a more robust solution by lifting 2D elements into 3D and leveraging Blender for precise manipulation. We let the diffusion model be a compositor for refinement and fusion, thus refraining it from understanding complex 3D and physical rules.

**Procedural Generation with Graphics Software.** Some recent works (Huang et al., 2024; Gu et al., 2025; Huang et al., 2025; Hu et al., 2024b) utilize pre-trained vision-language models (VLMs) to generate Blender Python code from user prompts. They execute synthesized code in Blender to render the image. Our framework is loosely connected to procedural generation in the sense that we can programmatically edit the projected 3D elements within Blender. However, instead of relying on a VLM to interpret user intent, we empower users to directly and interactively manipulate the elements as they desire. The resulting render is then passed to our compositor for the final image.

## 3    BLENDERFUSION

The goal of this work is to enable precise, 3D-aware visual compositing by integrating generative models with a symbolic graphics tool, such as Blender, across the three fundamental steps of compositing: layering, editing, and compositing. Concretely, we obtain editable 3D object entities from multiple images (layering), import them into Blender for versatile 3D-grounded scene modifications (editing), and use a diffusion-based visual compositor to convert coarse Blender renders and a background into realistic final images (compositing). Figure 2 illustrates the full training pipeline.

A core component is the visual compositor. Since the reconstructed 3D scene $S^{\text{src}}$ is derived from 2D images, its transformation often introduces noise, leading to artifacts in the target render $R^{\text{tgt}}$. The compositor corrects the artifacts with learned 3D shape priors to produce photorealistic outputs. Effective training requires paired scenes before and after editing. Object-centric videos naturally provide such supervision, but typically entangle object and camera motion, which limits generalization and disentangled control at test time. This motivates two simple yet effective training strategies, introduced in §3.2. We begin in §3.1 by presenting the overall pipeline, including object segmentation and 3D lifting, scene editing in Blender, and the architecture of the diffusion-based compositor.

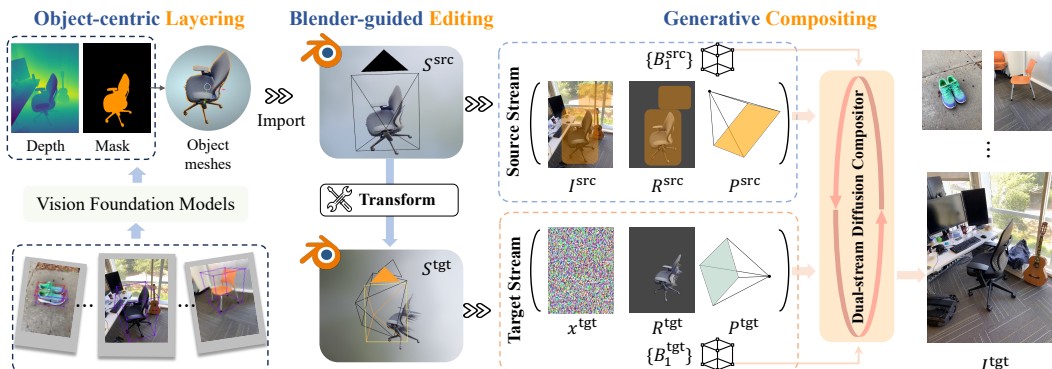

Figure 2: Pipeline of BlenderFusion. We simulate test-time transformations in Blender with video frames. The text embeddings of each stream encode the category label and 3D box of objects. The source masking strategy (the overlaid orange bounding boxes in $I^{\text{src}}$ and $R^{\text{src}}$) is detailed in §3.2.

## 3.1 BLENDERFUSION PIPELINE

**Object-centric Layering.** The layering step leverages off-the-shelf vision foundation models to obtain editable object reconstructions from input images. The framework supports multiple input images, each of which contains one or more objects. For generality, we assume that each object is reconstructed from a single image, while multi-view inputs can produce better reconstructions with recent 3D foundation models (Wang et al., 2024; Leroy et al., 2024; Wang et al., 2025).

We begin by projecting 3D boxes into the image plane to obtain coarse 2D boxes. The projected 2D boxes are usually loose, and we refine them with Grounding DINO (Liu et al., 2024) by prompting it with object categories. In practice, if the predicted box has a high overlap with the projected box (IoU > 0.5), we replace the latter. With the refined 2D box, we prompt SAM2 (Ravi et al., 2024) to extract the object mask. Combining this mask with metric depth predictions from Depth Pro (Bochkovskii et al., 2024), we obtain object-wise depth. We then adjust the scale of the predicted depth to align with each object's 3D bounding box. Finally, we back-project adjusted object-wise depth maps to generate 3D point clouds and connect adjacent points to form a triangle mesh. This results in a set of 3D entities, $S^{\text{src}}$, which are imported into Blender for subsequent editing.

Note that the layering process described above is efficient and applied to all the data. At test time, more advanced layering operations can be included for higher-quality reconstructions. When the editing task requires a complete and high-quality 3D model (e.g., complicated intra-object part-level editing, or material change, etc.), we optionally use image-to-3D models (Xiang et al., 2024; Zhao et al., 2025) to produce complete object meshes. Please see §B for details.

**Blender-guided Editing.** The output of the layering step, $S^{\text{src}}$, is then imported into Blender or other graphics software, where versatile transforms can be applied to objects and the camera with precise 3D grounding. BlenderFusion covers the following control tasks:

• ***Basic object control*** includes the translation, rotation, or scaling of each independent object, as well as object removal, insertion, or replacement. Since $S^{\text{src}}$ contains per-object 3D models, all these transformations can be applied automatically and reflected in the render $R^{\text{tgt}}$.

• ***Advanced object control*** represents object attribute change (e.g., color, material), non-rigid object transforms (e.g., part-level control, deformation, etc.), and novel object insertion (e.g., not covered by the training data). These operations are inherited from Blender and can be fulfilled by user interactions in the user interface or automatic scripts (Figure 1 and Figure 8).

• ***Camera and Background control*** involves camera motion and background replacement. The new background is specified with an image to replace $I^{\text{src}}$, while the camera motion is simulated through the camera object in $S^{\text{src}}$.

After applying all these edits, both the original and edited scenes, $S^{\text{src}}$ and $S^{\text{tgt}}$, are rendered into $R^{\text{src}}$ and $R^{\text{tgt}}$, providing reliable 3D-grounded control signals for the compositing step. $R^{\text{src}}$ and $R^{\text{tgt}}$ contain the rendered RGB image and an object index mask from Blender's Object Index Pass. Although the training data only covers simple object transformations and camera motion, our generative compositor generalizes to all the above editing tasks at test time.

**Generative Compositing.** The generative compositor fuses two streams of information (Figure 2). The *source stream* consists of the image $I^{\text{src}}$, its render $R^{\text{src}}$, its camera parameters $P^{\text{src}}$ and object poses $B^{\text{src}}$. The *target stream* includes the render of the edited scene $R^{\text{tgt}}$, its camera parameters $P^{\text{tgt}}$, and the target object poses $B^{\text{tgt}}$. $R^{\text{tgt}}$ is noisy due to transforming imperfect reconstructions. To effectively process the dual-stream input, we adapt a pre-trained diffusion model (e.g., a UNet or Diffusion Transformer (Peebles & Xie, 2023)) with three simple and general modifications:

1) We extend the model to a dual-stream architecture following the dual-stream input, where a single weight-shared network processes both streams independently while enabling cross-stream interaction through self-attention (Shi et al., 2023; Tang et al., 2024; Gao et al., 2024).

2) We modify the first layer of the model to accommodate additional conditions, increasing its input channels from 4 to 15 with zero-initialized weights. The original 4 channels are the VAE-encoded image or noise for the source or target stream. 5 additional channels handle Blender renders (4 for the VAE-encoded render and 1 for the instance mask). The remaining 6 channels encode camera parameters with Plücker embeddings (Sitzmann et al., 2021).

3) Each object has a class label and a 3D box. The label is CLIP-embedded (Radford et al., 2021) while 3D boxes are positional encoded (Vaswani et al., 2017) and processed by an MLP. The embeddings of each object are concatenated into a sequence, serving as the text tokens for each stream.

## 3.2 GENERATIVE COMPOSITOR TRAINING

Figure 2 (right) presents the generative compositor. To simulate test-time editing, we sample a source (original) and a target (edited) frame from a video. $I^{\text{src}}$ goes through the layering step to obtain $S^{\text{src}}$ and $R^{\text{src}}$. $S^{\text{src}}$ is then transformed with the 3D object bounding boxes (object pose) and camera parameters (view changes) to obtain $S^{\text{tgt}}$, and re-rendered into $R^{\text{tgt}}$. This process introduces noise into the target render within the reconstruction, alignment, and transformation process. Given the noisy target render, the model is trained to complete missing object textures and geometry, filling in the view-changed background, using the source stream as context. While effective, this training strategy has two key limitations. First, the model struggles with edits that intensively modify the original context, such as object removal, insertion, or background change. Second, it performs poorly with disentangled object control, especially when the camera remains fixed. To this end, we introduce our two specific training strategies.

**Source Masking.** If the original context is altered (e.g., an object is removed, replaced, or aggressively edited), the model should disregard the modified region in the source when compositing the target image. To achieve this, we introduce the source masking strategy during training, where each object in $I^{\text{src}}$ and $R^{\text{src}}$ is randomly masked out with a probability of 0.5 (see the overlaid orange boxes in Figure 2). At test time, we then flexibly mask both the source image and source render, or only mask the source image, depending on the concrete control task. This training strategy also has a regularization effect, which mitigates the potential over-reliance on source information and relative camera pose, and enforces the model to more accurately follow the intended edits in the target Blender render (Figure 7). We also apply random masking to background regions to prevent the model from getting an inpainting bias. Please refer to §B.1 for complete details.

**Simulated Object Jittering.** We observe that object motion supervision in training videos is limited and often strongly entangled with camera motion. For example, in Objectron videos, objects usually remain static while only the camera moves. To alleviate this issue, we introduce an object jittering training strategy that simulates dynamic object motions under a fixed-camera setup, as shown in Figure 3. The key change from the standard video learning setup is the replacement of $I^{\text{src}}$ and $P^{\text{src}}$ with $I^{\text{tgt}}$ and $P^{\text{tgt}}$. A random source masking strategy is applied separately to $I^{\text{tgt}}$ and $R^{\text{src}}$. To

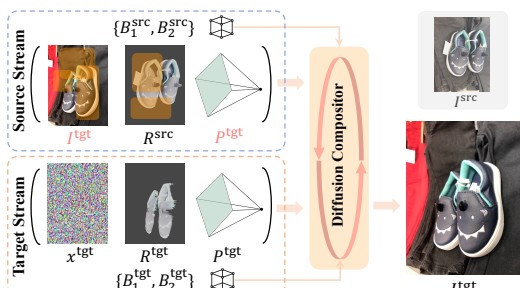

Figure 3: The simulated object jittering strategy for improving disentangled object control.

infer the masked $I^{\text{tgt}}$ from the noisy $R^{\text{tgt}}$, the model learns to effectively leverage the object information in $R^{\text{src}}$, $B^{\text{src}}$, and $B^{\text{tgt}}$, while the camera motion remains fixed. Despite its simplicity, this approach provides accurate, disentangled control of the object and camera at test time.

## 4 EXPERIMENTS

In this section, §4.1 describes the implementation details and experimental setups. §4.2 and §4.3 then compare the approaches on 3D-aware control tasks of increasing complexity and finer granularity. Finally, §4.4 presents additional results and analyses. §A provides demo/interactive results.

### 4.1 EXPERIMENTAL SETTINGS

The diffusion compositor is fine-tuned from Stable Diffusion v2.1 and is trained with 8 NVIDIA A100 80GB GPUs. The batch size is 320, and the model is trained for 30,000 iterations. We use the AdamW optimizer with a weight decay factor of 1e-2 and a 500-step linear warmup. The learning rate is 5e-5 for the diffusion model and 1e-4 for the MLP that encodes 3D boxes. For inference, we run DDPM (Ho et al., 2020) sampler for 50 steps, and use classifier-free guidance (CFG) (Ho & Salimans, 2022) with scale 2.0. When encoding the camera parameters, the camera of the source stream sets the reference coordinate system. The 3D bounding box is projected into the image plane, and each corner is represented by (x, y, depth). We then introduce datasets, baselines, and metrics. Please refer to §B and §C of the Appendix for complete implementation details.

**Datasets.** We cover three public video datasets with 3D object and camera annotations:

• *MOVi-E* is a synthetic multi-object video dataset from the Kubric (Greff et al., 2022). To obtain videos with more objects/camera dynamics for a more challenging evaluation, we run the official data generation code to produce 10,000 MOVi-E videos by increasing the number of dynamic objects and the range of camera motion. The image resolution is $512{\times}512$.

• *Objectron (Ahmadyan et al., 2021)* contains 15,000 real-world object-centric video clips across 9 categories, recorded with camera movement in real-world environments. Note that this dataset only contains static objects. We keep the aspect ratio of the dataset and resize the images to $384{\times}512$.

• *Waymo Open Dataset (WOD) (Sun et al., 2020)* consists of 1,000 real-world videos captured from self-driving cars. Following prior work (Wu et al., 2024), we use the front-view camera and filter out small cars. We keep the aspect ratio of the original dataset and resize the images to $528{\times}352$.

**Baselines.** We use 3DIT and NA as baselines. Image Sculpting is excluded as it needs per-scene optimization and mainly handles single objects. Full details of baseline adaptations are in §C.2.

• *Object 3DIT (Michel et al., 2024)* originally adapts Zero-1-to-3 (Liu et al., 2023) to incorporate a text instruction describing the control task and is trained on simple synthetic data, mainly targeting single-object control. We introduce several updates for fair comparisons. The base model is replaced with Stable Diffusion (SD) v2.1, and the source image is encoded by the SD VAE and concatenated with the input, following the original approach. Additionally, the Plücker embedding of the relative camera pose is concatenated. Instead of using a plain text instruction, we replace it with serialized object embeddings to enable multi-object control. Each object embedding consists of the CLIP embedding of its object category and the encoding of its 3D bounding box.

• *Neural Assets (NA) (Wu et al., 2024)* enables multi-object 3D editing through object tokens that encode appearance and pose features. The object appearance is obtained by applying RoIAlign (He et al., 2017) to DINO features of the foreground image, while the pose is obtained from the 3D object bounding box with an MLP. The background is processed separately, with the appearance extracted from the foreground-masked DINO features and the pose computed from the relative camera pose between the source and target images. Tokens for all objects and the background are serialized into a sequence, replacing the text embedding in the pre-trained SD v2.1 architecture.

**Evaluation Metrics.** We follow the evaluation metrics used by NA. PSNR, SSIM, and LPIPS are computed at image and object levels. The FID and per-object DINO feature cosine similarity are also reported. See §C.3 for details. All quantitative evaluations follow the standard video setup, which measures how well the model transforms a source frame into a target frame given camera and object poses. Note that this setup cannot assess fine-grained, disentangled control tasks (without ground truth answers), which are more relevant and practical in real-world editing scenarios. Thus, we conduct qualitative comparisons and human evaluations to better evaluate these capabilities.

**Synthetic Data Pre-training.** We observe that all approaches struggle to learn disentangled object rotation on WOD, primarily due to the lack of angular object motion (i.e., cars mostly exhibit simple translation across frames). To address this, we initialize WOD training with the pre-trained check-

Table 2: Quantitative results with MOVi-E (Greff et al., 2022), Objectron (Ahmadyan et al., 2021), and Waymo Open Dataset (WOD) (Sun et al., 2020), using the standard video setup. The model controls both the objects and the camera to transform the source frame into the target frame.

| Dataset | Model | Object-level metrics | | | | Frame-level metrics | | | |
|---|---|---|---|---|---|---|---|---|---|
| | | PSNR ↑ | SSIM ↑ | LPIPS ↓ | DINO ↑ | PSNR ↑ | SSIM ↑ | LPIPS ↓ | FID ↓ |
| MOVi-E | Object 3DIT | 14.06 | 0.284 | 0.411 | 0.848 | 17.02 | 0.500 | 0.340 | 15.71 |
| | Neural Assets | 13.74 | 0.221 | 0.428 | 0.826 | 16.73 | 0.414 | 0.388 | 23.08 |
| | BlenderFusion | 18.90 | 0.557 | 0.227 | 0.914 | 21.32 | 0.674 | 0.198 | 9.11 |
| Objectron | Object 3DIT | 13.88 | 0.290 | 0.424 | 0.902 | 14.98 | 0.355 | 0.423 | 6.14 |
| | Neural Assets | 13.73 | 0.278 | 0.427 | 0.921 | 14.56 | 0.337 | 0.427 | 6.18 |
| | BlenderFusion | 16.06 | 0.389 | 0.291 | 0.959 | 16.54 | 0.413 | 0.323 | 3.25 |
| WOD | Object 3DIT | 18.90 | 0.448 | 0.255 | 0.930 | 23.21 | 0.640 | 0.220 | 11.92 |
| | Neural Assets | 16.87 | 0.301 | 0.322 | 0.901 | 20.41 | 0.548 | 0.267 | 15.39 |
| | BlenderFusion | 20.93 | 0.596 | 0.185 | 0.956 | 24.11 | 0.676 | 0.189 | 10.02 |

point from MOVi-E, which contains richer object motions and provides stronger 3D object priors. For all three approaches, we adopt this initialization and reduce the base learning rate to $1e-5$. See §C.1 for dataset details, and §D.3 for failure cases on WOD without synthetic pre-training.

## 4.2 STANDARD EVALUATION WITH VIDEO FRAME SETUP

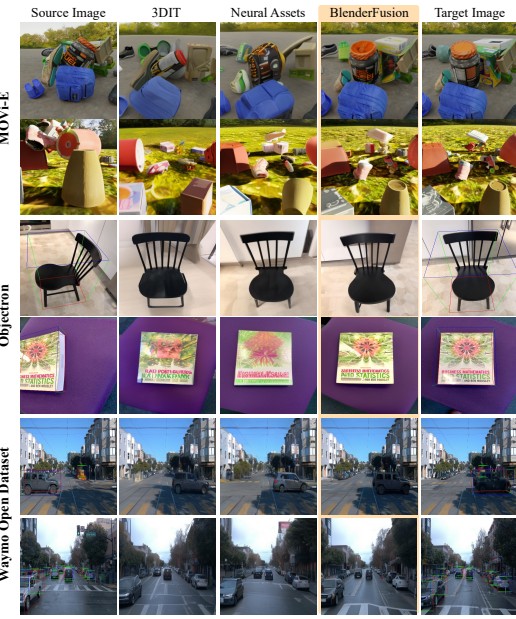

Figure 4: Comparisons with video frame setup.

Table 2 and Figure 4 present results of the standard video setup. Quantitatively, BlenderFusion consistently improves object and image-level metrics across all datasets, indicating better modeling of foreground and background. Qualitatively, we preserve the precise appearance and identity of objects while accurately capturing their geometry and shading. In contrast, baselines often distort details (e.g., the chair back in the third row, the car in the fifth row). Additionally, the baselines struggle to handle a large number of dynamic objects with various transforms in MOVi-E, whereas ours demonstrates reliable generation quality across multiple objects. However, this evaluation setup *entangles camera and object dynamics* and is unsuitable for evaluating disentangled control. For example, a model entirely overfitted to camera motion achieves superior performance on Objectron. At test time, a critical use case is to keep the camera frozen while manipulating the objects and background (§4.3).

## 4.3 DISENTANGLED CONTROL AND FINE-GRAINED COMPOSITING

**Disentangled Control.** Figure 5 studies disentangled controls over objects and camera. While 3DIT performs reasonably well in the standard video setup, it fails on all disentangled object manipulation tasks. It tends to keep the object still, showing a strong entanglement of object and camera motion. NA demonstrates clearly better results, but has two limitations: 1) It loses appearance and geometric details because its DINO encoder does lossy encoding and likely discards fine details even after fine-tuning; 2) Foreground and background interfere with each other, which arises because RoIAlign on DINO features cannot clearly separate objects from the background. In contrast, BlenderFusion precisely follows the editing intent while preserving both geometry and appearance details, thanks to the reliable 3D grounding from the Blender renders. Please check §A for interactive results.

**Fine-grained Editing and Compositing.** Figure 6 further explores more complex 3D-aware compositional editing tasks. As task complexity increases, the benefits of explicit 3D grounding become more pronounced. Concretely, NA fails to handle multiple small objects and does not preserve ob-

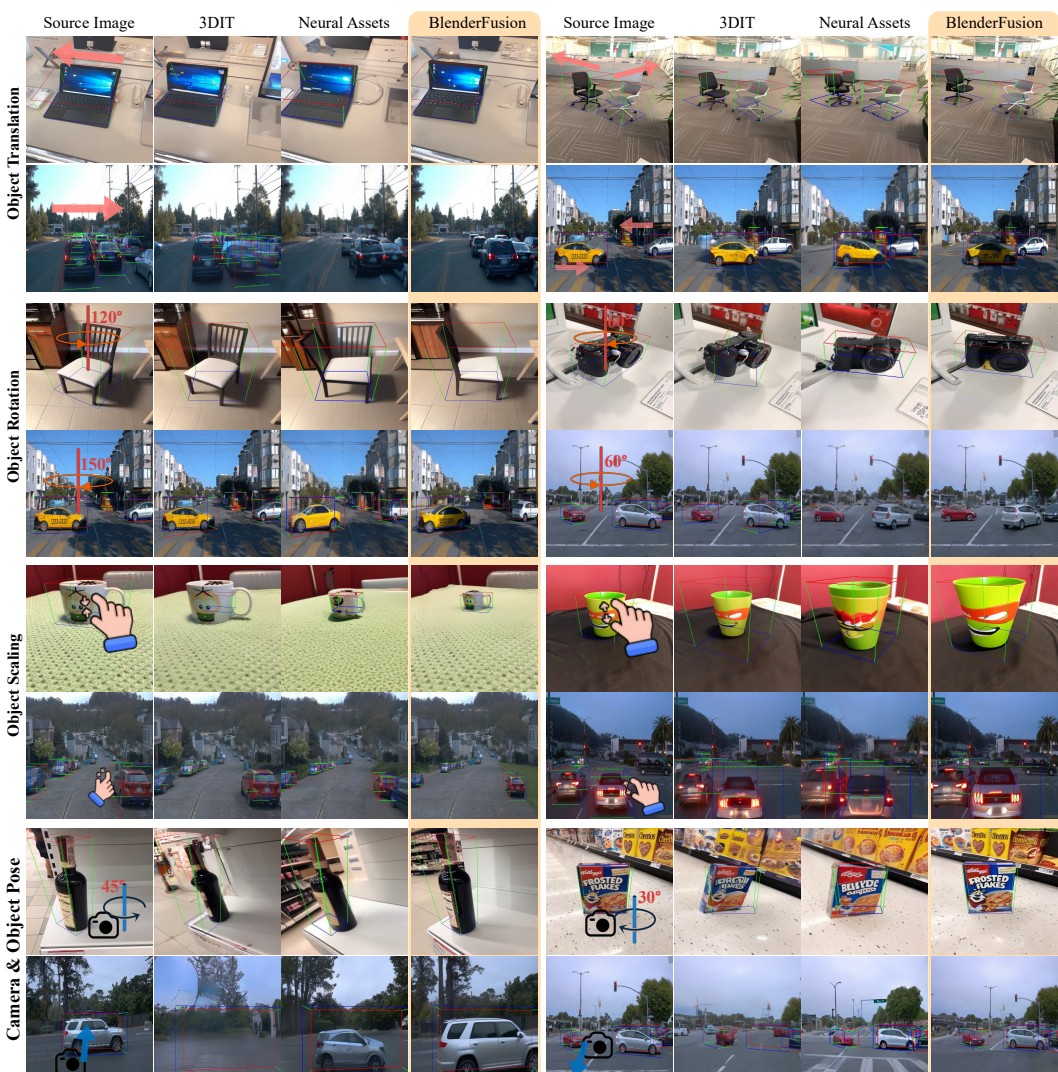

Figure 5: Qualitative comparison for disentangled control of object and camera on Objectron and WOD. BlenderFusion demonstrates more precise control and more consistent object identity.

ject identity. On the contrary, our method easily handles the duplication of 5 small cups (row 1), consistently retains both the appearance and geometry of objects (row 2 and 3), and produces more realistic shading (row 3). Ours also preserves the background much better than NA. These results demonstrate the impact of our core idea–decoupling control from generation, leveraging Blender as a bridge to achieve precise and flexible visual compositing. Figure 12 provides more examples.

## 4.4 MORE RESULTS

**Human Evaluation.** Based on the three evaluation settings in §4.2 and §4.3, we conduct a user study to further validate the superiority of our method. We curate 54 examples for human evaluation: 18 for the standard video frame setup, 24 for disentangled object control, and

| Setting | Ours (%) | Baseline (%) | Draw (%) |
|---|---|---|---|
| Overall | 87.04 | 6.40 | 6.56 |
| Video | 80.79 | 8.80 | 10.42 |
| Disentangled | 88.37 | 6.60 | 5.03 |
| Fine-grained | 93.75 | 2.43 | 3.82 |

12 for complex fine-grained compositional control. Participants compared two shuffled generations (ours vs. Neural Assets) and selected one: A, B, or Similar. The results were collected from 1,294 selections from 24 users and are summarized in the right table. The gap between BlenderFusion and the baseline gets larger when the editing task becomes more challenging, which further demonstrates the advantages of the 3D-grounded framework in complex and compositional visual editing.

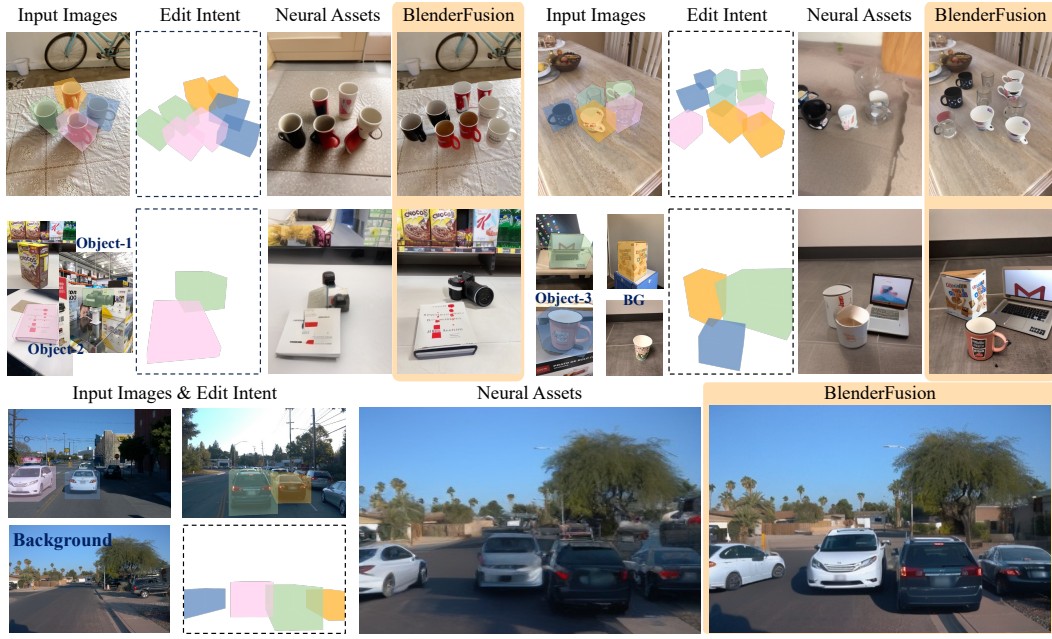

Figure 6: Qualitative results on fine-grained compositing tasks. The edit intent presents the desired scene geometry while the color encodes the object identity. More results are in Figure 12.

**Ablation Studies.** Figure 7 investigates the 3DIT baseline, a dual-stream model variant without Blender, and three design choices based on our Blender control interface. Specifically, the third column utilizes our dual-stream model, incorporating the depth and segmentation of the input image (from DepthPro and SAM2) into the source stream, while having no Blender renders. This variant only shows improvements over the 3DIT baseline in the object translation setting. When adding the Blender renders, the dual-stream model can do object translation, but the background moves along with the object. It also struggles with disentan-

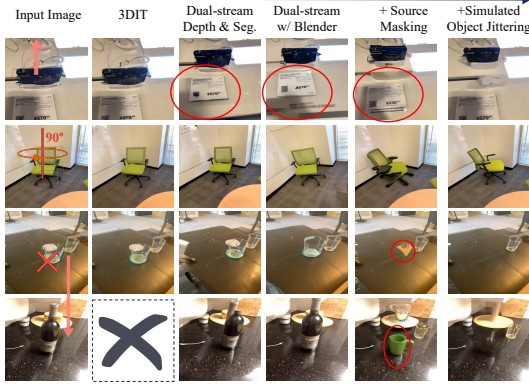

Figure 7: Qualitative ablation results.

gled object rotation and instance-wise removal, and fails in background changes. The source masking improves object control, but the model still suffers from entangled camera and object states. This is mainly because the training data rarely covers object manipulation under a fixed camera. The simulated object jittering strategy addresses these remaining issues, significantly enhancing the disentangled control capabilities of both object and background. Note that the source masking and simulated object jittering are designed based on the Blender renders. Both training strategies become invalid without the $R^{\text{src}}$ and $R^{\text{tgt}}$ in the compositor's inputs. Please refer to the Appendix for more results: §A provides video demos and interactive results; §D presents generalization results, quantitative ablation studies, failure case analyses, and more qualitative examples.

## 5 CONCLUSION

This paper introduces BlenderFusion, a novel framework that integrates image generation models with 3D graphics tools, enabling fine-grained and highly controllable visual editing. BlenderFusion follows three key steps: it segments and lifts objects from 2D images into editable 3D entities, performs precise manipulations on objects and camera within Blender, and refines the resulting renders into a final image through our diffusion model compositor. Experiments on both synthetic and real-word datasets demonstrate that BlenderFusion significantly outperforms prior works, providing a flexible and practical framework for 3D-aware visual editing and composition.

**Reproducibility Statement.** All datasets (MOVi-E, Objectron, and Waymo Open Dataset), pre-trained models (Stable Diffusion, SAM2, Depth Pro, etc.), and software (Blender, Diffusers) used in the paper are publicly available resources. The paper provides complete details of datasets, baselines, model implementations, and evaluation metrics in §4.1, §B, and §C.

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

# Appendix

## A    VIDEO DEMO AND INTERACTIVE LOCAL WEBPAGE

To better demonstrate BlenderFusion's core ideas and its capabilities, we provide two additional resources: 1) a video that explains our layering-editing-compositing pipeline with demos for each step, and presents the method's capabilities in disentangled object/camera control and fine-grained multi-image compositing; 2) a local webpage from which the readers can check more results **interactively** and compare with the main baseline (Neural Assets). We explain the details below.

### A.1    VIDEO DEMO

Please first unzip the supplementary material zip file and open the file **demo_video.mp4**. This video provides a brief introduction to our method's three steps, accompanied by qualitative results for disentangled object/camera control and more complex visual editing and composition tasks.

### A.2    LOCAL WEB PAGE WITH INTERACTIVE RESULTS

Please first unzip the supplementary material zip file and open the folder **local_page**. Then, click **index.html** to open our local web page. This page provides the following information:

⬦ A thorough motivation of the paper, introducing typical failure cases with state-of-the-art commercial image editing models on 3D-aware visual editing and compositing.

⬦ An overview of our method with animations to help understand the three core steps.

⬦ **Interactive results** for disentangled object and camera control. Please follow the instructions on the page to view the results of our method and baselines.

⬦ A compact re-compilation of the key qualitative results shown in our main paper and appendix.

## B    REMAINING IMPLEMENTATION DETAILS OF BLENDERFUSION

The entire framework of BlenderFusion is implemented with Diffusers (von Platen et al., 2022), Pytorch 2.6, Blender 3.6.9 and its Python APIs (BPY). The diffusion compositor's training and inference are with bfloat16 to save GPU memory cost.

### B.1    TRAINING DETAILS

When training the generative compositor (the diffusion model adapted from Stable Diffusion v2.1), we use the v-prediction diffusion training objective (Salimans & Ho, 2022). We enable gradient checkpointing, use gradient accumulation with 2 accumulation steps, and train the model with mixed bfloat16 precision. The ratios of vanilla video training, training with source masking, training with both source masking and simulated object jittering, and unconditional training (for CFG) are 0.35, 0.3, 0.3, and 0.05, respectively.

#### B.1.1    SOURCE MASKING

During training, the source masking is always consistently applied to the source image and the source Blender render. Concretely, we randomly mask each foreground object with an (independent) probability of 0.5, then apply random masking to the background using boxes with a similar aspect ratio as the corresponding foreground objects. The mask is applied by masking out the 2D bounding box derived from the 3D object bounding box, with a dilation operation. The background masking here effectively alleviates the object inpainting bias, which might hurt the performance in object removal. When an object is masked out, the corresponding source bounding box information is also dropped for the source stream. The simulated object jittering training also applies the same source masking strategy, while the masks applied to the "source" image and source render are not the same—because the "source" image in this reconstruction setting is actually the target frame, while the source render still comes from the true source frame.

### B.1.2  FRAME SAMPLING STRATEGY FOR TRAINING DATA

To train BlenderFusion's generative compositor, we need to prepare the Blender renderings of the source and target frames. For both datasets, we sample the source frame uniformly with a fixed stride. Then, for each source frame, we randomly sample a set of target frames. For each pair of source and target frames, we obtain the object 3D models with the 3D lifting process, import them into Blender, and simulate the object and camera transforms. In this way, we get the source and target renders for training our dual-stream diffusion model.

Note that since the two baselines do not rely on the reconstruction and re-render steps, there is no limitation on the sampling of the source and target frames, and they actually train with more diverse data than BlenderFusion. Therefore, the strong quantitative and qualitative performance of our method suggests that training with stronger 3D grounding can also improve the data efficiency.

### B.2  TEST-TIME DETAILS

### B.2.1  TEST-TIME LAYERING DETAILS

At test time, the layering step only uses the 2.5D surface reconstructions without running image-to-3D models for complete meshes, except for the complex editing tasks in Figure 8 (Bottom).

When using an image-to-3D model, we do the following steps to obtain the object mesh. Concretely, we crop out the image patch of each object using the SAM2 mask and run Hunyuan3D v2 (Zhao et al., 2025) with the cropped image to obtain a complete textured mesh. We then align the mesh with the object's 3D box and the 2.5D surface reconstruction.

### B.2.2  TEST-TIME COMPOSITOR DETAILS

For the standard video evaluation setting, all conditions are directly passed to the model without any masking or dropping. For disentangled object manipulation with fixed camera (including translation, rotation, and scaling), the source masking is applied to two regions in the *source image*: 1) the region of the original object and 2) the expected region of the target object. The *source render* and the *source object 3D bounding box* are only masked/dropped if the object is intended to be removed or replaced, otherwise, they are preserved and will be used as the main reference for object appearance and geometry. The source masking at test time encourages better disentanglement between foreground objects and background contexts.

When applying the advanced object-level control inherited from Blender (e.g., attribute change, deformation), as described in §3.1, those edits are always reflected in both the source render $R^{\mathrm{src}}$ and target render $R^{\mathrm{tgt}}$ in the dual-stream inputs. Concretely, when we close the screen of the laptop like in Figure 1, the edit is always first reflected in the initial scene $S^{\mathrm{src}}$ and then rendered to update $R^{\mathrm{src}}$, then, further edits like object rotations or translations transform $S^{\mathrm{src}}$ into $S^{\mathrm{tgt}}$ and we render $S^{\mathrm{tgt}}$ to get $R^{\mathrm{tgt}}$. In this way, the difference between the target and source stream is always the basic object transformations, which is consistent with the training setting and makes the framework generalize well to much more complex multi-object editing and scene composition tasks at test time.

## C  REMAINING DETAILS OF DATASETS, BASELINES, AND EVALUATION

### C.1  DATASET DETAILS

**MOVi-E Details.** Instead of using the original 10K MOVi-E videos released by the Kubric dataset generator (Greff et al., 2022), we update the generation config to generate a new set of 10K videos with more extensive object and camera motions. Specifically, we update the number of static objects from 10-20 to 5-10, and the number of dynamic objects from 1-3 to 5-10. For the camera movement, we reset the maximum camera movement from 4 units in Kubric's 3D space to 8 units. This MOVi-E variant provides a challenging benchmark for multi-object control with 3D awareness (occlusion, novel viewpoint, etc.). It is thus more suitable for evaluating the model's 3D controllability over multiple objects and the camera. Additionally, it makes the synthetic MOVi-E data a more effective pre-training resource for real-world datasets with limited object and camera motions (e.g., WOD has highly imbalanced object poses).

**Objectron Details.** Similar to NA's setting, we drop the bike category and use the remaining data for training and evaluation. For each video, we randomly sample 60 frames without repeating to get a clip. Then, we randomly sample source and target frames for each clip. The test set contains 2812 videos, and we sample 67,092 pairs for evaluation. The original resolution of the dataset is $480\times640$ (height is larger than the width). We keep the aspect ratio and use $384\times512$.

**Waymo Open Dataset (WOD) Details.** Similar to the settings of NA, we only take the front-view camera of WOD, and use the same filtering strategies to filter out the videos without large objects. Concretely, the pair of source and target frames is considered as valid only when there is at least one large object that occupies more than 1% image area in both frames. We sample 6976 pairs of source and target frames from the test set for evaluation.

## C.2 BASELINE DETAILS

For controlled experiments between baselines and ours, we re-implement 3DIT and NA within the Diffusers framework, strictly following the descriptions in their original papers. We ensure all approaches use the same base model (i.e., Stable Diffusion v2.1), the same raw input information (input image, object category, source/target object 3D box), identical training setups (training loss, training iterations, learning rate, etc.), and identical sampling setups (sampler and sampler steps). As a result, the primary difference lies in their core controllability strategy. Concretely, 3DIT directly uses the text embeddings (i.e., class labels and source/target 3D boxes) to transform the source image; NA employs external DINO (Caron et al., 2021) to extract the per-object appearance and combines it with target 3D box, then controls through the text embedding interface; BlenderFusion leverages a more explicit interface, Blender, to obtain source and target renders as 3D-grounded control signals, where many fine-grained 3D visual edits are possible.

Note that the re-implemented 3DIT uses the same input image resolution as ours, and shares the same training recipes and inference setups. It can be considered as ours without the two-stream model architecture, the Blender renders, and the two accompanying training strategies.

In our experiments, we observe that training NA with our generation resolutions (i.e., $384\times512$ for Objectron, $528\times352$ for WOD) produces apparent object appearance shifts compared to NA's original $256\times256$ setting used, making both the qualitative and quantitative results worse. This might be related to the resolution inconsistency between its DINO encoder (i.e., $224\times224$) and the diffusion model, and is not trivially resolved by jointly fine-tuning all modules. Increasing DINO's input resolution significantly raises the training GPU memory cost and makes the training unaffordable, which also has the risk of invalidating the pre-trained priors. Therefore, for better qualitative results under DINO's pre-defined input resolution, we keep NA's original $256\times256$ generation resolution in all relevant experiments.

## C.3 EVALUATION DETAILS

Our quantitative evaluation metrics follow those used in Neural Assets (Wu et al., 2024). For object-level metrics (PSNR, SSIM, LPIPS, and DINO), we use the 2D object box to crop out the local image patch and compute the metrics between the generated and ground-truth patches. The 2D object box is derived by projecting the ground-truth 3D object box into 2D using the provided camera parameters.

All the metric implementations are based on the torchmetrics library (Nicki Skafte Detlefsen et al., 2022). All generated and ground-truth images are resized to $256\times256$ before computing the quantitative metrics to align with NA's output resolution and evaluation settings.

# D ADDITIONAL EXPERIMENTAL RESULTS

## D.1 GENERALIZATION RESULTS

### D.1.1 GENERALIZATION TO UNSEEN DATASETS

We apply BlenderFusion trained on Objectron data to in-the-wild images from SUN-RGBD (Song et al., 2015), ARKitScenes (Baruch et al., 2021), and Hypersim (Roberts et al., 2021) datasets.

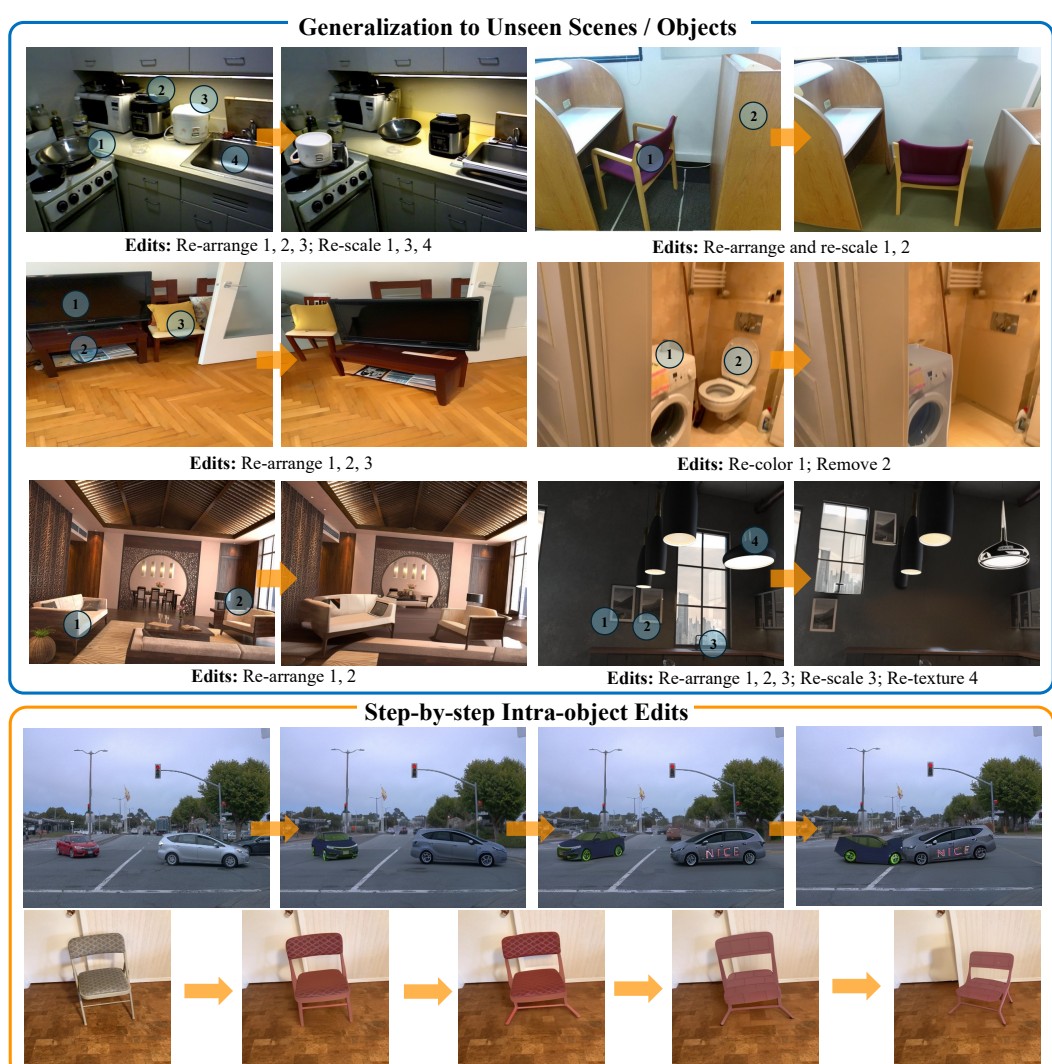

Figure 8: (Top) BlenderFusion shows can generalize to unseen images from SUN-RGBD Song et al. (2015), ARKitScenes Baruch et al. (2021), and Hypersim Roberts et al. (2021) datasets. (Bottom) Our framework inherits the versatile editing capabilities of graphics software, enabling diverse object control tasks beyond the scope of the training data. Images are resized to facilitate visualization.

All three datasets present scenes with much higher complexity and richer details than Objectron. Figure 8 (Top) demonstrates the generalization results. Those in-the-wild images present much more complicated scene structure and object details than the Objectron training data, and Hypersim is a high-end synthetic dataset created by professional designers. BlenderFusion presents reasonable generalization capabilities, although the visual quality shows some degradation compared to in-domain results.

### D.1.2 GENERALIZATION TO INTRA-OBJECT PROGRESSIVE EDITING

In Figure 8 (Bottom), we demonstrate progressive, step-by-step editing, where each intermediate modification in Blender is rendered using our diffusion compositor. For these examples, Hunyuan3D v2 is utilized during the layering step to generate higher-quality 3D object models, as detailed in §3.1. The first row illustrates: 1) color changes for each object, 2) rotation and text engraving, and 3) object deformation. The second row shows: 1) color change, 2) part-level deformation, 3) texture replacement, and 4) rotation. Although we present simple, progressive edits here, more advanced controls supported by Blender are also possible.

## D.2  Additional Ablation Study

Figure 7 of the main paper presents the qualitative ablation study. A straightforward ablation model variant is to use our dual-stream design, but directly employ the depth and segmentation of the input image from Depth Pro and SAM2 in the source stream (third column). Note that this variant does not use any Blender renders, and the target stream only contains the noisy latents and the camera embeddings.

The other three variants study the key components of our method: 1) the dual-stream architecture, 2) the source masking training, and 3) the simulated object jittering training. As discussed earlier, those disentangled control tasks are more suitable for evaluating the model's practical editing capabilities compared to the standard video frame evaluation setup. However, there is a lack of standard evaluation metrics as no ground truth target image is available.

Table 3: The quantitative ablation studies on the key elements of our diffusion model compositor, corresponding to Figure 7 of the main paper.

| Method | Object-level Metrics | | | |
|---|---|---|---|---|
| | PSNR↑ | SSIM↑ | LPIPS↓ | FID↓ |
| 3DIT (one-stream) | 13.88 | 0.290 | 0.424 | 6.14 |
| Dual-stream (DS) | 15.90 | 0.378 | 0.310 | 3.52 |
| DS + Depth & Seg. | 16.04 | 0.382 | 0.313 | 3.74 |
| DS + Blender | 16.05 | 0.389 | 0.292 | 2.93 |
| + Source Masking | 16.18 | 0.393 | 0.290 | 2.64 |
| + Sim. Obj Jittering | 16.06 | 0.389 | 0.291 | 3.25 |

To make the ablation study more complete, we provide the quantitative ablation results with the *standard video setting* in Table 3, using the same model variants as Figure 7. The dual-stream design alone can improve the baseline significantly by more effectively leveraging the source and target information. Incorporating the Blender renders in both streams further unlocks the model's capacity by providing reliable 3D groundings. However, as already presented in Figure 7, these variants still cannot achieve disentangled object control, struggling to manipulate objects with a fixed camera or to do significant modifications to the source image (e.g., object removal, background change). The source masking can slightly improve the quantitative results, as it can be considered as a data augmentation strategy to the basic video training setting. While being the core design to acquire disentangled object control, the simulated object jittering training strategy slightly lowers the quantitative results on the standard video setting. This is expected because it is essentially an image reconstruction training, and has a different source stream setting from the standard video setup (i.e., source camera is the same as the target camera).

## D.3  Failure Cases and Analyses

In our experiments, one main failure mode is that the model sometimes has difficulty achieving accurate manipulation for disentangled object rotation (Figure 9). On WOD, this is largely alleviated by using the MOVi-E pretrained model as the initialization to compensate for the lack of pose difference between the source and target frames in the training data (as cars keep going straight in most of the time). The top row of Figure 9 shows typical failure cases of NA and BlenderFusion when not using MOVI-E pre-training. Notet that this type

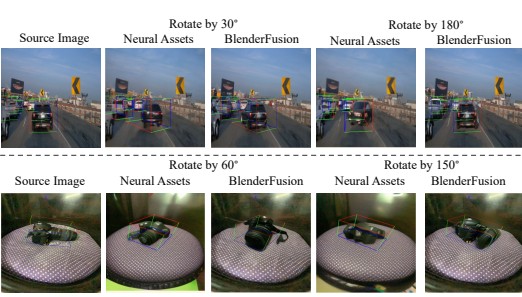

Figure 9: Failure cases on object rotation.

of failure case is *significantly alleviated for both NA and BlenderFusion with MOVI-E pre-training*. This result highlights that creating high-quality synthetic data can be a promising direction to improve 3D-aware complicated visual editing and control.

On Objectron, the failure of object rotation usually comes with wrong or obviously altered object geometry, and is caused by two factors: 1) the model lacks accurate 3D understanding for complex object geometry (e.g., cameras with complicated structures), and 2) when the object reconstruction is 2.5D, the renders can be unreliable when the object is rotated significantly. For the latter case, using 3D-Gen meshes resolves the problem in most cases, albeit at the cost of a longer processing time for running an image-to-3D model and an object pose alignment step. Different from the results on WOD, pre-training on MOVi-E does not help with this failure case, probably because the objects in MOVi-E only present simple geometry. Two potential directions can be explored in future works: 1) pre-training on datasets with diverse object geometries and motions. For example, creating synthetic

data using the diverse Objaverse dataset (Deitke et al., 2023); or 2) extending the pipeline to multi-view or video input so that more complete 3D reconstructions can be obtained with state-of-the-art multi-view reconstruction approaches, such as VGGT (Wang et al., 2025).

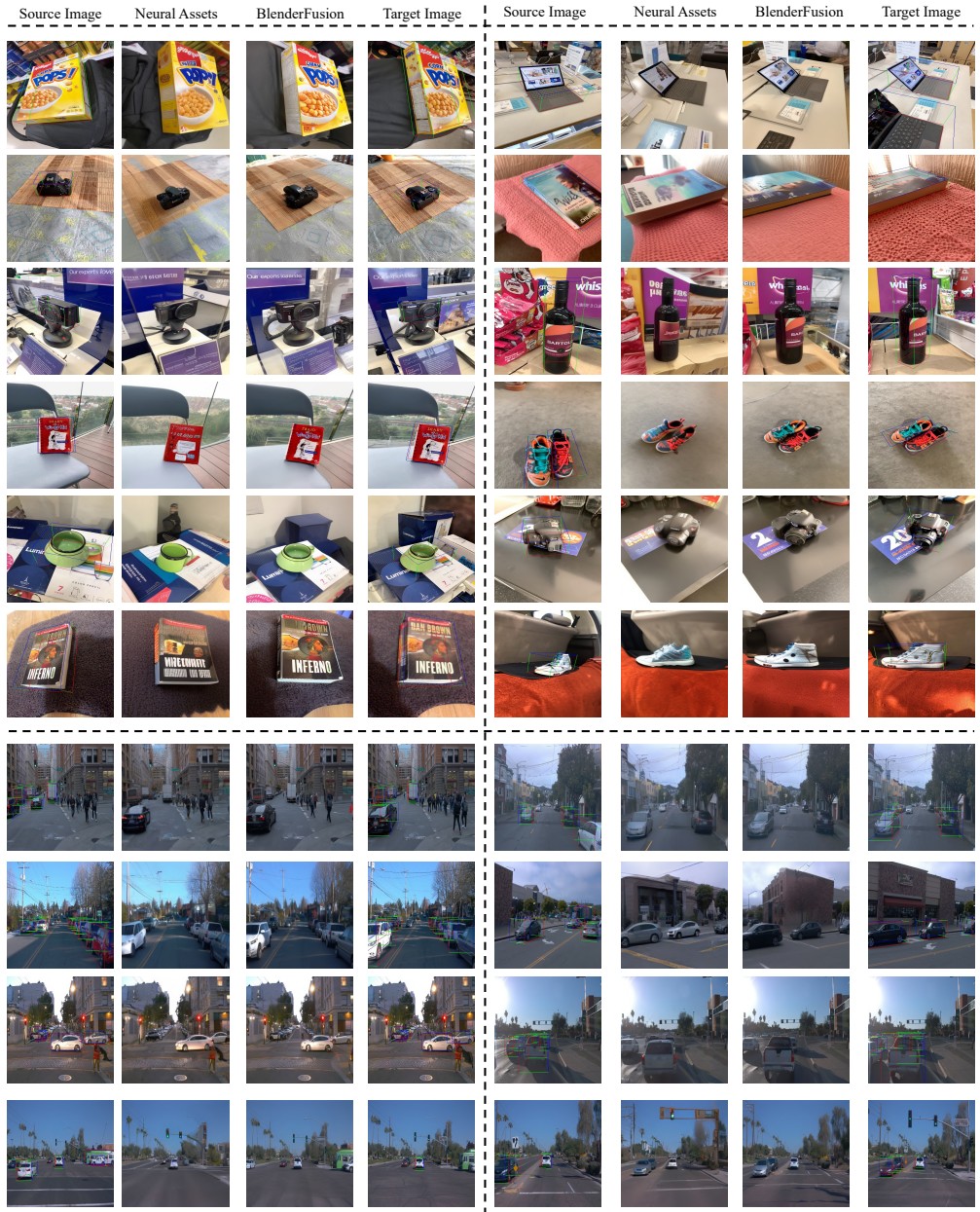

Figure 10: Additional qualitative results of the standard evaluation setting (source and target frames from a video), extending Figure 4 of the main paper. 3DIT is omitted in this figure.

### D.4 ADDITIONAL QUALITATIVE RESULTS

Figure 10, Figure 11, and Figure 12 provide additional qualitative comparison, extending Figure 4, Figure 5, and Figure 6 of the main paper. These examples further demonstrate the superior disentangled control and compositional editing capability of BlenderFusion.

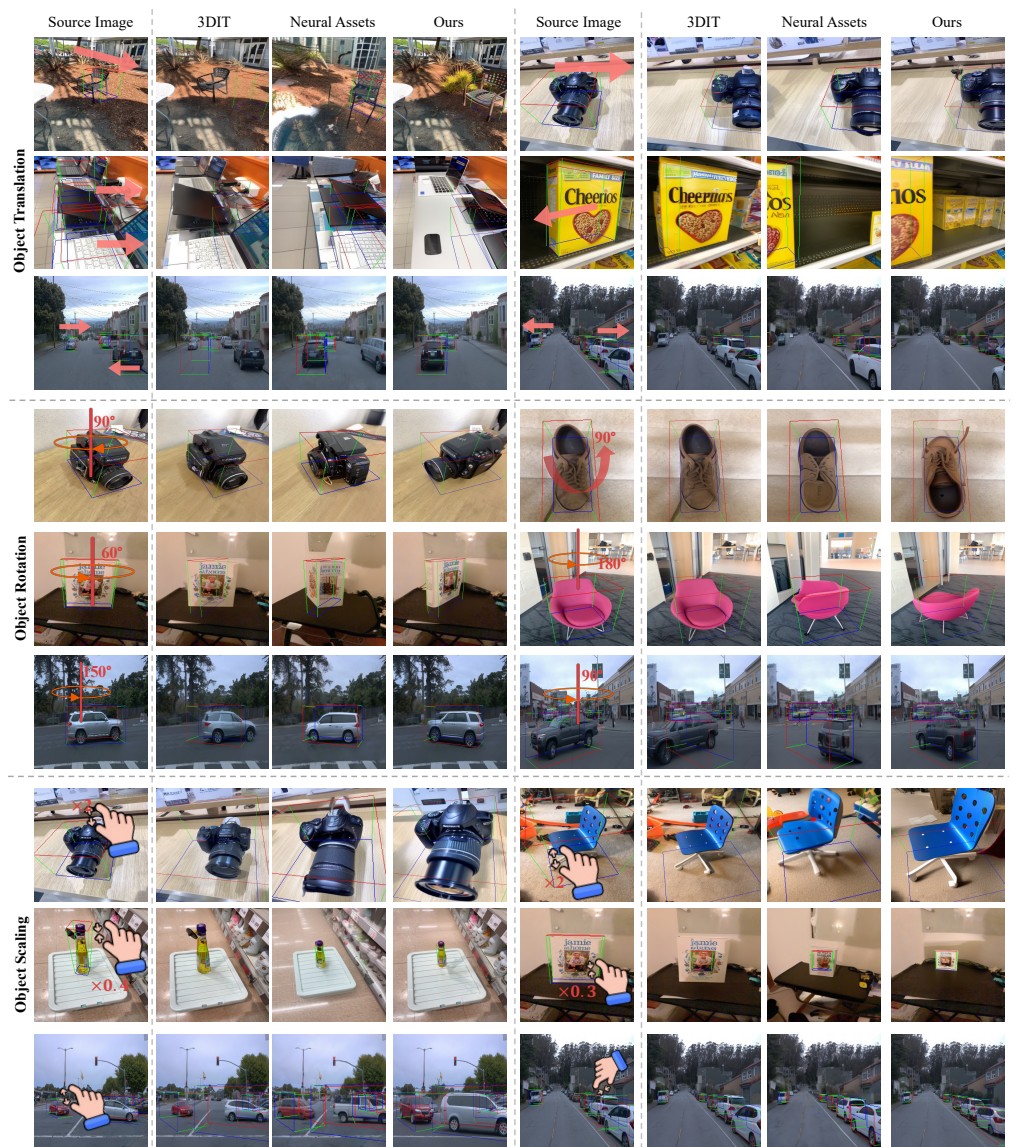

Figure 11: Additional qualitative results of disentangled object control with fixed camera, extending Figure 5 of the main paper.

# E    THE USE OF LARGE LANGUAGE MODELS

Large Language Models (LLMs) are used to polish the paper's writing, adjust LaTeX formats, and assist with the development of the local webpage in the supplementary material (A.2). LLMs are not used in the method design process and core model implementations.

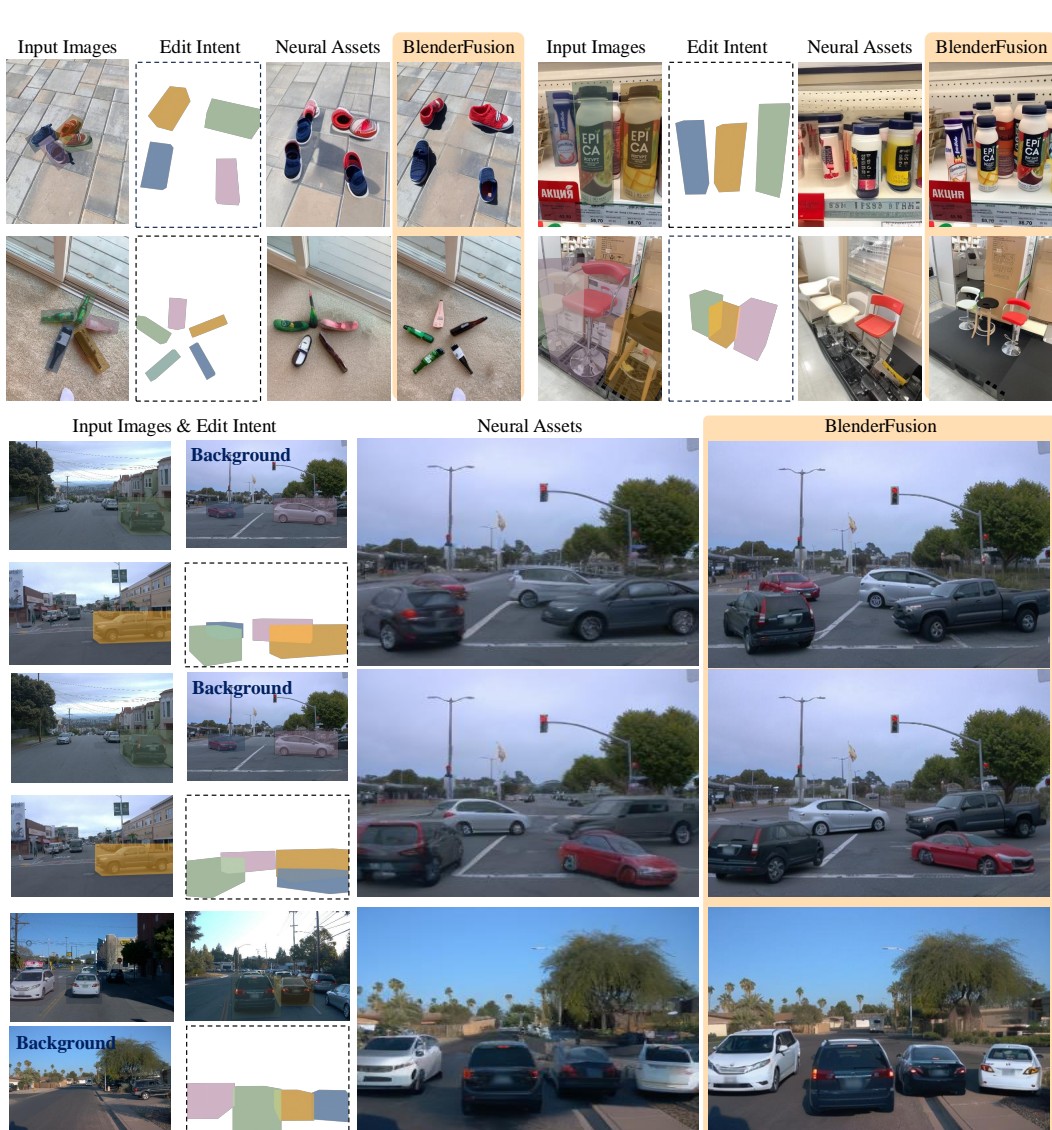

Figure 12: Additional qualitative results on fine-grained multi-object editing and compositing tasks, extending Figure 6 of the main paper. The edit intent demonstrates the expected output geometry while the color encodes the object identity.

## F    ADDITIONAL RESULTS ADDED DURING REBUTTAL PERIOD

The results in this section are prepared for the rebuttal. We created this separate section to make cross-references easier during the rebuttal period. We will incorporate the contents into the paper in future revisions.

### F.1    ARCHITECTURE DETAILS OF THE DIFFUSION COMPOSITOR

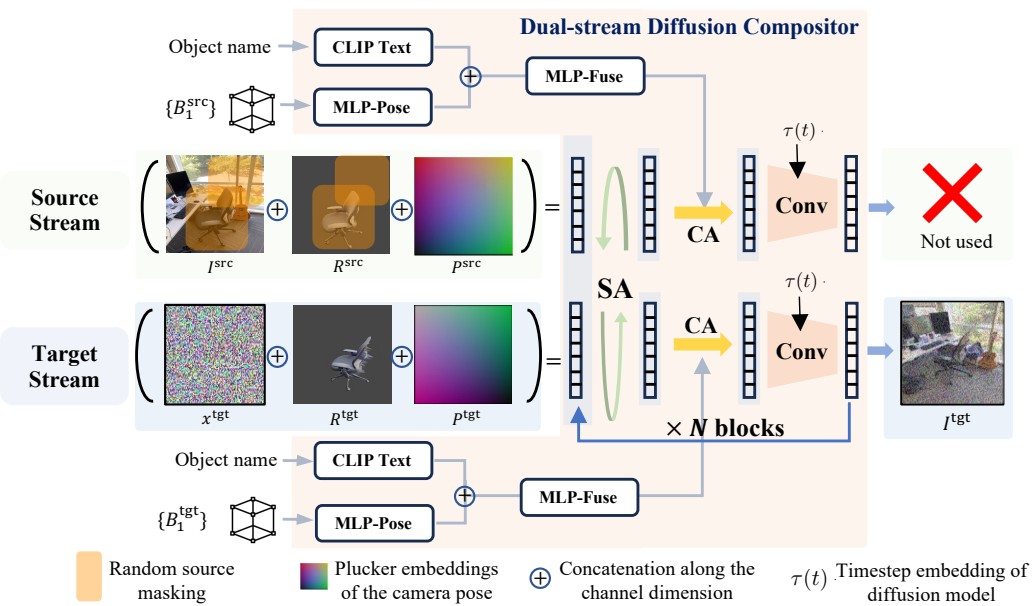

Figure 13: The detailed architecture of the dual-stream diffusion compositor. Both real images and Blender renders are encoded using the VAE encoder of the base diffusion model (Stable Diffusion v2.1). "SA" and "CA" denote the self-attention and cross-attention layers within each network block of the base model. Note that the first convolution layer of the pre-trained diffusion model is extended to accommodate additional input channels.

Figure 13 illustrates the full architectural details of our dual-stream diffusion compositor.

### F.2    COMPARISON WITH DIFFUSION HANDLES

Figure 14 shows the results of Diffusion Handles (Pandey et al., 2024) under our basic experimental settings of Table 2 and Figure 4. It is a training-free method that manipulates a pre-trained depth-to-image diffusion model with depth warping and feature injection. The results become extremely bad when the camera motion is large, and the number of objects increases.

### F.3    COMPLICATED RESULTS WITH MORE THAN 10 OBJECTS

Figure 15 presents additional examples from the MOVI-E dataset. Each example contains more than 10 objects, with substantial changes in object poses and camera viewpoints. Our method clearly outperforms the baselines in terms of object's geometric correctness, appearance preservation, and camera control accuracy.

### F.4    COMPARISON AGAINST ZEROCOMP ON DISENTANGLED OBJECT CONTROL

Figure 16 shows results from an additional baseline, ZeroComp (Zhang et al., 2025), following the reviewer's suggestion. Since ZeroComp does not support camera motion, we evaluate it only on the basic disentangled object-control tasks.

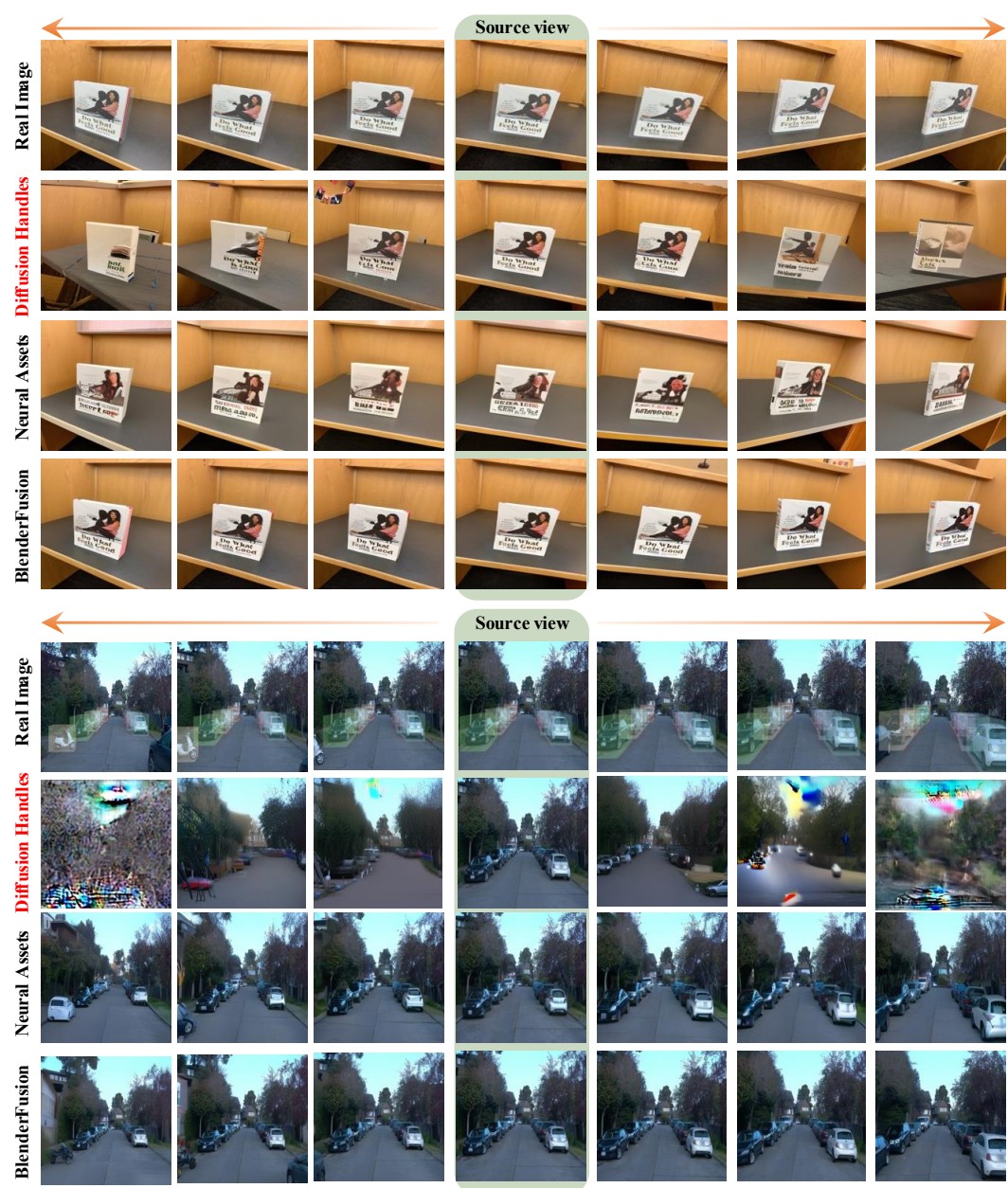

Figure 14: Results of Diffusion Handles (Pandey et al., 2024) under our experimental setup with multi-object editing and camera viewpoint change. The top example is from Objectron, and the bottom is from Waymo Open Dataset. The hard depth warping makes it struggle with tasks beyond single-object editing and static viewoint.

To apply ZeroComp in our setting, we generate per-object meshes using the image-to-3D model Hunyuan 3D 2.0 and align them to the input image using the available 3D bounding boxes and our 2.5D rough mesh (the same procedure used in Figure 8, bottom). We also perform background inpainting, as ZeroComp requires a clean background image as input.

As shown in the figure, ZeroComp has difficulty with real-world images: it does not properly handle lighting and shading, and its performance is highly sensitive to errors in the input 3D meshes. Inaccuracies from monocular 3D lifting propagate directly to the final composite, resulting in severe artifacts, particularly on Waymo examples. This aligns with the fact that ZeroComp is trained entirely on synthetic data with fully controlled intrinsic layers and assumes clean foreground–background

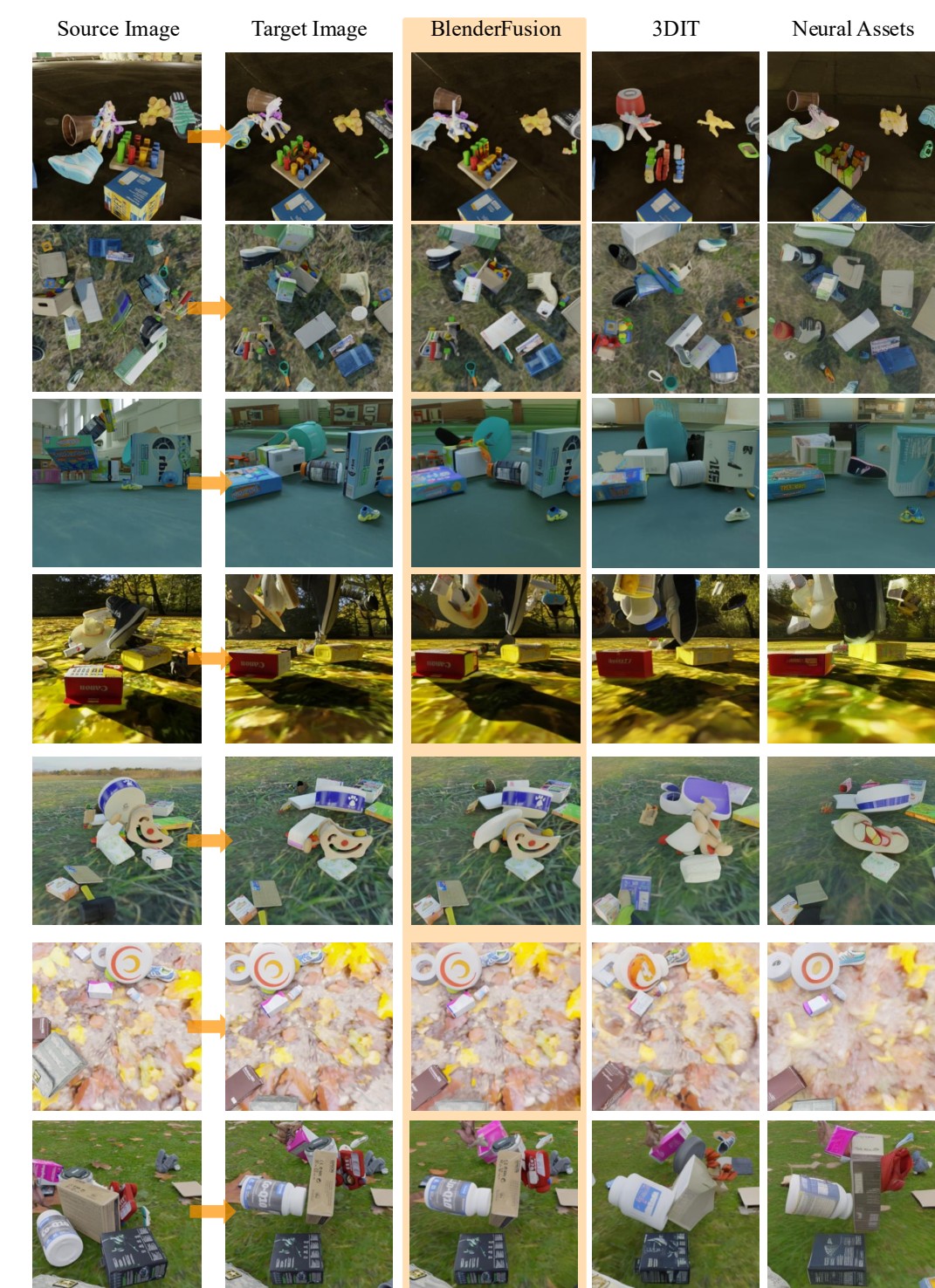

Figure 15: Additional results on the MoVI-E dataset, focusing on scenes containing **over 10 objects**. Between the source and target images, objects experience **substantial 3D pose variations alongside notable changes in camera viewpoint**. Under these complex re-composition tasks, BlenderFusion clearly surpasses 3DIT and Neural Assets in maintaining geometric consistency and object identity.

separation, conditions that do not hold for real images. Moreover, its training pipeline is not directly applicable to real-world data without additional handling of FG–BG separation.

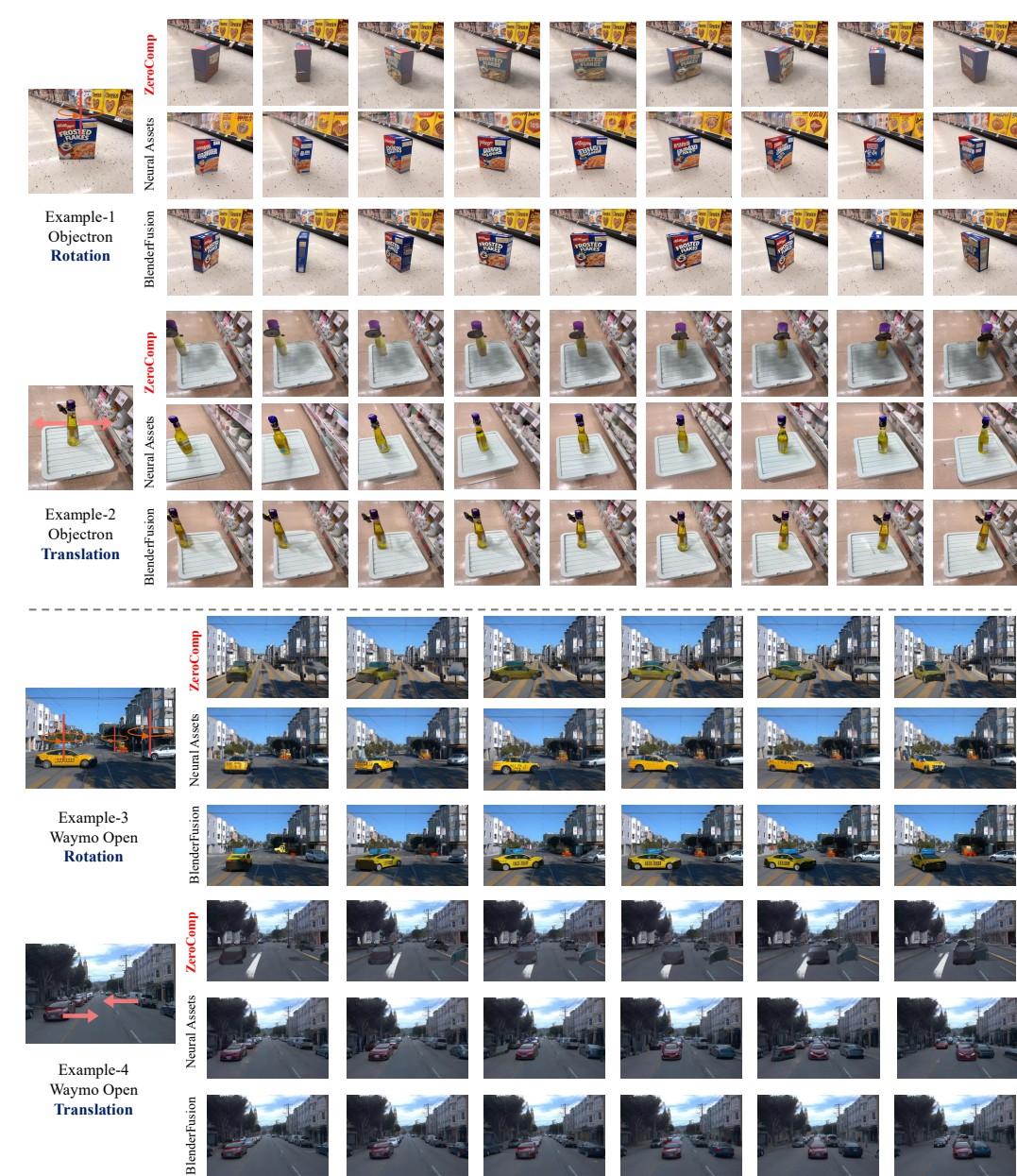

Figure 16: Results of ZeroComp (Zhang et al., 2025) on Objectron and Waymo Open Dataset.

In summary, although ZeroComp is conceptually related through its use of Blender and diffusion, it addresses a different problem: inserting clean, complete 3D assets with a focus on lighting and shading correction. In contrast, our tasks require editing and recomposing objects directly from real images without assuming ground-truth meshes and must also handle arbitrary camera pose changes. Under these conditions, ZeroComp does not produce results comparable to 3DIT or Neural Assets, and is therefore not an appropriate baseline.

## F.5 ADDITIONAL EVALUATION FOR BACKGROUND AND FOREGROUND FIDELITY

**Background PSNR/SSIM.** We report background fidelity (pixels outside the foreground bounding boxes) in Table 4.

Table 4: Background PSNR/SSIM across datasets.

| Method | MoVI-E | | Objectron | | Waymo | |
|---|---|---|---|---|---|---|
| | PSNR | SSIM | PSNR | SSIM | PSNR | SSIM |
| 3DIT | $25.42 \pm 5.52$ | $0.82 \pm 0.10$ | $18.17 \pm 4.20$ | $0.62 \pm 0.15$ | $24.23 \pm 4.27$ | $0.69 \pm 0.11$ |
| Neural Assets | $24.49 \pm 4.86$ | $0.75 \pm 0.11$ | $17.50 \pm 3.79$ | $0.59 \pm 0.16$ | $22.46 \pm 3.68$ | $0.61 \pm 0.10$ |
| BlenderFusion | $\mathbf{29.78 \pm 6.07}$ | $\mathbf{0.88 \pm 0.07}$ | $\mathbf{19.78 \pm 4.81}$ | $\mathbf{0.69 \pm 0.15}$ | $\mathbf{25.19 \pm 4.50}$ | $\mathbf{0.72 \pm 0.10}$ |

**DINO/CLIP similarity (inserted asset vs composite).** Table 5 presents the DINO and CLIP similarity for the disentangled control and compositing tasks (the style of Figure 5 and Figure 6). For each object, features from the source bounding box are compared to those in the composited image.

Table 5: Foreground appearance similarity (rounded).

| Method | Objectron | | Waymo | |
|---|---|---|---|---|
| | DINO | CLIP | DINO | CLIP |
| Neural Assets | $0.748 \pm 0.085$ | $0.717 \pm 0.098$ | $0.840 \pm 0.057$ | $0.771 \pm 0.054$ |
| BlenderFusion | $\mathbf{0.866 \pm 0.052}$ | $\mathbf{0.810 \pm 0.072}$ | $\mathbf{0.891 \pm 0.059}$ | $\mathbf{0.822 \pm 0.050}$ |

## F.6 TRAINING COST AND INFERENCE SPEED

**Training cost.** Using the setup described in Sec. 4.1, the training time of each method on the Objectron dataset (using eight NVIDIA A100 80GB GPUs) is approximately:

- **3DIT:** ~30 hours.
- **Neural Assets:** ~40 hours. This method trains the DINO encoder jointly with the diffusion model and performs region-level feature extraction for every object, resulting in higher cost than 3DIT.
- **BlenderFusion (diffusion compositor):** ~48 hours. Training the dual-stream architecture takes longer than single-stream models such as 3DIT and Neural Assets.

The layering and Blender editing components are not trained.

**Inference speed.** We measure inference time on the Objectron dataset using the same configuration as Table 2 and Fig. 4: batch size of 12, 50 DDPM sampling steps, and a single NVIDIA A100 80GB GPU.

- **3DIT:** 6.16 s (resolution $384 \times 512$).
- **Neural Assets:** 5.94 s at $256 \times 256$ and 10.92 s at $512 \times 512$. As discussed in the submission, we report NA's results at $256 \times 256$ (the setting used in its original paper), because both quantitative and qualitative performance degrade at higher resolution, likely due to the DINO encoder being pre-trained at $224 \times 224$.
- **BlenderFusion:**
  - Layering (SAM2.1 + Depth Pro + back-projection to obtain 2.5D meshes): 0.6 s. These foundation models are highly efficient.
  - Blender scripting for manipulating the 12 scenes in the batch: 3.0 s. The 2.5D proxy meshes are lightweight, allowing the Blender Python scripts to run quickly without requiring GPU rendering.
  - Diffusion compositor: 9.48 s (resolution $384 \times 512$).

  **Total:** 13.08 s.

