# OpenReview forum: "BlenderFusion: 3D-Grounded Visual Editing and Generative Compositing"
_ICLR.cc/2026/Conference — Submitted to ICLR 2026_

### Official Review · Reviewer_k2sd · 2025-10-16

**Soundness:** 3
**Presentation:** 3
**Contribution:** 2
**Rating:** 4
**Confidence:** 4

**Summary:**

This paper proposes BlenderFusion, a framework for 3D-aware multi-object image generation. From the perspective of controllable generation, this work enables controlling the 3D pose and appearance of objects in an image. From the perspective of tool use in foundation models, this work leverages Blender as a physical engine to generate 3D consistent images. Compared to prior works like Object 3DIT and Neural Assets (NA), BlenderFusion explicitly reconstructs the 3D model of an object, transforms it in the 3D space, and then renders a photorealistic image by fine-tuning Stable Diffusion. By training on paired images or video frames, it is able to disentangle the pose and appearance of objects, enabling various editing applications. Both qualitative and quantitative results on three datasets show that BlenderFusion clearly outperforms baselines.

**Strengths:**

- The paper is well-motivated. It largely resembles the traditional workflow of computer graphics -- first create 3D assets (including geometry and appearance), then port them into software like Blender, and finally render to images.
- The results on this specific task are clearly better than baselines, especially the performance under extreme conditions like composing a large number of objects or large rotation and translation.

**Weaknesses:**

1. The novelty seems limited. Nothing is surprising to me after reading the paper:
- The object-centric modeling approach is similar to Neural Assets, which represents an object with 3D pose and appearance;
- Using explicit 3D representation or tool use to improve visual generative models has been explored extensively [1, 2].

2. My biggest concern is that this approach is likely not scalable. It reminds me of a line of work in neural-symbolic learning which converts objects to abstract representations, and then runs external tools like code or physical engine to simulate the world [3, 4, 5]. These methods often have a strong assumption about the perception models, e.g., they should be robust enough to reconstruct the 3D geometry of objects from in-the-wild images.
- Though object reconstruction from object-centric images is to some extent solved, objects in real-world images are often not well-defined. For example, the part-whole hierarchy often causes ambiguity. Severe object occlusion will also degrade the perception model's performance.

3. In some result videos on the attached website, it seems that background pixels also change when foreground objects are edited.
4. What's the training cost and inference speed of BlenderFusion compared to baselines like NA?

[1] Hu, Ziniu, et al. "Scenecraft: An llm agent for synthesizing 3d scenes as blender code." Forty-first International Conference on Machine Learning. 2024.

[2] Ren, Xuanchi, et al. "Gen3c: 3d-informed world-consistent video generation with precise camera control." Proceedings of the Computer Vision and Pattern Recognition Conference. 2025.

[3] Ding, Mingyu, et al. "Dynamic visual reasoning by learning differentiable physics models from video and language." Advances in Neural Information Processing Systems 34 (2021): 887-899.

[4] Rubanova, Yulia, et al. "Learning rigid-body simulators over implicit shapes for large-scale scenes and vision." Advances in Neural Information Processing Systems 37 (2024): 125809-125838.

[5] Tang, Hao, Darren Key, and Kevin Ellis. "Worldcoder, a model-based llm agent: Building world models by writing code and interacting with the environment." Advances in Neural Information Processing Systems 37 (2024): 70148-70212.

**Questions:**

1. I wonder how important Blender is in this 3D-aware editing task. A prior work Diffusion Handles [1] uses depth to warp diffusion features of an object to target location, and also fine-tunes a diffusion model to render images. Can you compare with one such baseline to show the advantage of Blender?
2. I'm a bit confused by the "Simulated Object Jittering" technique. In this case, the source image is replaced with the target image as the model input. Why won't the model just use its information to denoise the target image? How can the model learn anything in this case?

[1] Pandey, Karran, et al. "Diffusion handles enabling 3d edits for diffusion models by lifting activations to 3d." Proceedings of the IEEE/CVF Conference on Computer Vision and Pattern Recognition. 2024.

---

> ### Author Response · Authors · 2025-11-24
> **Response to Reviewer k2sd (1/3)**
>
> We thank the reviewer for the insightful questions. We try to address the concerns and questions one by one below.
>
> ---
> ### **1. Novelty and technical contribution**
>
> We would like to clarify the technical contributions of BlenderFusion. We do not claim object-centric modeling itself as a novelty
>
> Instead, our goal is different: we show that combining a lightweight 3D editing proxy in Blender with a diffusion compositor trained to use it enables editing and compositing tasks that existing object-centric baselines cannot handle. Specifically, as demonstrated in Fig. 6, Fig. 8, and Fig. 12, BlenderFusion supports multi-object, multi-viewpoint, and more complex scene manipulations that 3DIT and Neural Assets are not designed for.
>
> Our main technical contributions lie in:
>
> (i) the **meta-framework design**, which unifies vision foundation models (for layering), Blender (for editing), and diffusion models (for compositing) into a cohesive pipeline;
>
> (ii) the **dual-stream diffusion compositor architecture and training strategies** that make effective use of noisy Blender renders. As detailed in our clarification on “simulated object jittering,” this approach solves the non-trivial challenge of achieving disentangled control with limited real data.
>
> Regarding prior work integrating graphics engines with generative models, our approach differs significantly: Unlike **Gen3C** (which uses Blender to render 3D points from camera trajectories) or **Scenecraft** (which renders LLM-generated code), BlenderFusion utilizes Blender as a general editing interface.  This allows for a wide range of scene-level transformations—multi-object control, camera motion, object insertion/removal, part-level edits, and optional high-quality mesh substitution. The diffusion compositor is trained to refine these rough renders into photorealistic results, enabling a broader level of controllability than previous 3D-conditioned methods.
>
> In summary, BlenderFusion is neither a new object-centric representation nor a simple variant of existing tool-use methods. It is a practical and extensible framework that combines a flexible 3D editing proxy with a diffusion model trained to use its noisy signals. This combination enables capabilities not supported by 3DIT, Neural Assets, or prior 3D-guided generative models.
>
> ----
> ### **2. Scalability of our framework**
>
> We thank the reviewer for this insightful comparison. However, there is a fundamental distinction between the neural-symbolic works cited (which rely on **hard constraints**) and BlenderFusion (which utilizes **soft generative guidance**).
>
> Unlike traditional neural-symbolic methods, where perception errors cause cascading failures in the simulation engine, our framework does not assume accurate 3D geometry. We treat the 2.5D layering results as *noisy, coarse proxies* rather than ground truth. Our diffusion compositor is explicitly trained to tolerate and correct artifacts arising from occlusion, segmentation ambiguity, and inaccurate depth.
>
> This robustness is empirically demonstrated by our results on real-world datasets (Objectron and Waymo) and complex scenes with more than 10 objects (the synthetic MoVi-E dataset, **Fig.15** of the updated PDF). These datasets contain the in-the-wild challenges the reviewer mentions. BlenderFusion consistently outperforms 3DIT and Neural Assets in these settings, proving it does not suffer from the brittleness of the baselines.
>
> Finally, our scalability is future-proofed by design. Since we rely on modular components, BlenderFusion automatically benefits from the rapid scaling of vision foundation models (e.g., SAM, Depth Anything) and generative backbones, allowing performance to improve without architectural changes. Also, using synthetic data for pre-training, as we explored with MoVi-E, can further improve generalization across object categories and viewpoints, alleviating the dependence on limited real-world training data.
>
> ---
> ### **3. Clarify “Simulated Object Jittering”**
>
> We clarify the confusion as follows. Simulated Object Jittering is always used together with the **Source Masking** strategy, as illustrated in Fig. 3. During training, the model receives:
>
> 1. The **background of the target image**, where all foreground (FG) objects are masked out by the source-masking mechanism;
> 2. The **source Blender render** (also with random source masking);
> 3. The **target Blender render**.
>
> Note that the target camera pose is identical to the source camera pose. Under this setup, the model **cannot** directly denoise or copy the target FG pixels because they are mostly masked out. To reconstruct the object, the model must rely on two Blender renders and the object pose information (fed through the text embedding) to infer its correct geometry and appearance; there is no shortcut through the real target image. This mechanism is crucial for encouraging proper **disentangled control** as demonstrated by our ablation studies (Figure 7 and Table 3).

---

> ### Author Response · Authors · 2025-11-24
> **Response to Reviewer k2sd (2/3)**
>
> ### **4. Compare with Diffusion Handles**
>
> We include qualitative results of Diffusion Handles in **Figure 14 of the updated PDF**. It fails badly when multiple objects are edited and the camera moves.
>
> Below, we clarify a few details about Diffusion Handles and explain why we adopt a Blender-based editing interface rather than depth-warping:
>
> 1. **Diffusion Handles is a training-free, single-image, and mostly single-object method.**
>
>     It operates on a pre-trained depth-to-image diffusion model and performs feature warping based on monocular depth. There is no fine-tuning of the base model, and the method is primarily designed for single-object edits under a static camera.
>
> 2. **Extending Diffusion Handles to our setting is fundamentally challenging.**
>
>     Depth-warping relies on accurate pixel-wise correspondences. In multi-object scenes or with camera motion, even small monocular depth errors can break these correspondences, leading to severe artifacts. In contrast, our diffusion compositor is trained to handle **noisy and incomplete** 2.5D reconstructions from the layering stage—it refines the coarse Blender renders and is robust to imperfect geometry, pose errors, and self-occlusions. Fig.14 shows representative failure cases of depth warping in multi-object, moving-camera scenarios.
>
>     Diffusion Handles is not designed for the multi-object, multi-viewpoint compositional tasks targeted by our method and by baselines such as Neural Assets. This is why it was not included as a baseline in the original submission.
>
> **Advantages of using Blender as the editing interface.**
>
> Blender provides a **structured and expressive interface** for specifying 3D edits, which is essential for the editing and compositing tasks studied in our paper. The reason for using Blender rather than depth warping for our framework is be summarized as follows:
>
> - **Expressiveness of edits.** Blender allows us to represent complex transformations—multi-object control, camera motion, part-level edits, and high-quality mesh substitution—as explicit 3D edits on the layered scene. The object assets can come from multiple images or from existing 3D assets. Many of our editing tasks (e.g., multi-image compositing in Fig. 1 and Fig. 6, or intra-object attribute/geometry edits in Fig. 8 bottom) cannot be expressed through depth-warping, which assumes one-to-one pixel correspondences.
>
> - **Making the framework robust to noisy geometry.** Diffusion Handles is inherently sensitive to depth inaccuracies because feature warping requires precise correspondences. Our diffusion compositor, by contrast, is *trained* to operate on noisy and incomplete Blender renders, learning to correct reconstruction artifacts. The programmatic protocols of Blender make it easy to prepare source and target renders for all the training data, making the approach resilient to the coarse 2.5D geometry produced by the layering stage.
>
> In summary, Blender serves as an effective and general interface for defining 3D-grounded edits, while the diffusion compositor learns to fuse the noisy edits into photorealistic results. Combining the two components enables complex multi-object, multi-viewpoint edits that are difficult to achieve with depth-warping alone.
>
> ----
>
> ### **5. Background consistency**
>
> We compute PSNR and SSIM specifically on the background regions (i.e., pixels outside the foreground object’s bounding box), following the same setup as Table 2 and Fig. 4. The results below show that BlenderFusion achieves substantially better background preservation than both baselines:
>
> |  | MoVI-E |  | Objectron |  | Waymo |  |
> | --- | --- | --- | --- | --- | --- | --- |
> |  | PSNR | SSIM | PSNR | SSIM | PSNR | SSIM |
> | 3DIT | 25.417 ± 5.515 | 0.817 ± 0.100 | 18.170 ± 4.197 | 0.622 ± 0.152 | 24.225 ± 4.269 | 0.687 ± 0.107 |
> | Neural Assets | 24.494 ± 4.858 | 0.753 ± 0.108 | 17.503 ± 3.793 | 0.592 ± 0.157 | 22.461 ± 3.683 | 0.609 ± 0.102 |
> | BlenderFusion | 29.777 ± 6.072 | 0.877 ± 0.074 | 19.776 ± 4.805 | 0.687 ± 0.145 | 25.187 ± 4.501| 0.723 ± 0.102 |
>
> The results confirm that our method maintains background fidelity with clear margins over the baselines. 3DIT works okay for background, but fails badly on disentangled control (Fig.5). Neural Assets suffer from the information loss of the DINO encoder.
>
> The remaining background inconsistencies seen in some videos occur mainly within the object bounding box. Because the source image is masked inside the bounding box when preparing inputs for the diffusion compositor, those pixels are re-synthesized rather than copied from the source image. Since each frame in the demo videos is generated independently (our method is image-based rather than video-based), the synthesized content in these regions may vary across frames, leading to the visible temporal differences. Improving temporal stability is an interesting direction for future work and may require using a video diffusion model as the base model.

---

> ### Author Response · Authors · 2025-11-24
> **Response to Reviewer k2sd (3/3)**
>
> ### **6. Training cost and inference speed**
>
> **Training cost.** Using the setup described in Sec. 4.1 of the main paper, the training time of each method on the Objectron dataset, using our eight NVIDIA A100 80G GPUs, is approximately:
>
> - **3DIT:** ~30 hours
> - **Neural Assets:** ~40 hours (it trains the DINO encoder together with the diffusion model, and needs to do region-level feature extraction for each object, thus takes a longer time than 3DIT)
> - **BlenderFusion (diffusion compositor):** ~48 hours (training the dual-stream model takes longer than a single-stream model like 3DIT and NA)
>
> The layering and Blender editing components are not trained.
>
> **Inference speed.** We compute inference time on the Objectron dataset using the same setup as in Table 2 and Fig. 4, with a batch size of 12 and 50 DDPM sampling steps. All the results are obtained with a single NVIDIA A100 80G GPU.
>
> - **3DIT:** 6.16 s (resolution 384x512)
> - **Neural Assets:** 5.94s (at resolution 256x256) and 10.92 s (at resolution 512x512).  As we described in the submission, we reported NA's results at 256x256 (the same as in its original paper), because we found that both the metric scores and qualitative results got worse at 512x512, likely because the DINO encoder was pre-trained at a low resolution (224x224).
> - **BlenderFusion:**
>     - Layering (SAM2.1 + Depth Pro + back-projeting to get 2.5D meshes): 0.6 s; these foundation models are quite efficient
>     - Blender scripting for manipulating 12 scenes of the batch: 3.0s. Our 2.5D proxy meshes are very lightweight, making the Blender Python scripts run very fast -- we don't even need to turn on GPU rendering for this.
>     - Diffusion compositor: 9.48 s (resolution 384x512)
>     - **Total:** 13.08 s

---

### Official Review · Reviewer_jWo3 · 2025-10-29

**Soundness:** 3
**Presentation:** 3
**Contribution:** 3
**Rating:** 8
**Confidence:** 4

**Summary:**

This paper presents BlenderFusion, a generative visual compositing framework that recomposes objects, camera, and background to synthesize novel scenes. The framework consists of three key stages: (i) segmentation and conversion of visual inputs into editable 3D entities (layering), (ii) manipulation in Blender with 3D-grounded control (editing), and (iii) fusion into a coherent scene using a generative compositor (compositing). Overall, the proposed method is well-motivated and the experimental results effectively support the claims.

**Strengths:**

•	The paper is well-written with a logical structure that makes the technical contributions easy to follow.
•	The proposed  framework is reasonable and well-justified. The experimental results convincingly demonstrate the effectiveness of the approach across various compositing scenarios.
•	Excellent supplementary materials: The demo videos and project page significantly aid in understanding the core concepts and practical applications of the method.

**Weaknesses:**

•	Recent works have explored 3D scene reconstruction and composition capabilities. A more thorough comparison and discussion of the relationship between BlenderFusion and these methods would strengthen the paper. For example:
•	CAST [1] performs component-aligned 3D scene reconstruction from a single RGB image. How does BlenderFusion's layering approach compare to CAST's decomposition strategy?
•	What are the trade-offs between the generative compositing approach and traditional 3D reconstruction-based composition methods?
•	Could you clarify when your method is preferable over existing 3D composition techniques?

Reference
[1] CAST: Component-Aligned 3D Scene Reconstruction From an RGB Image

**Questions:**

Could you pleasse examples about more complicated scenarios?

---

> ### Author Response · Authors · 2025-11-24
> **Response to Reviewer jWo3**
>
> We thank the reviewer for the insightful comments. We will address the questions one by one below.
>
> ----
>
> ### **1. Traditional 3D reconstruction-based composition methods**
>
> We thank the reviewer for bringing up CAST, and it is a very relevant work. Both CAST and our framework share a high-level philosophy of leveraging generative AI and vision foundation models to bridge the gap between 2D images and 3D manipulation. We discuss the similarities and differences in the following three aspects:
>
> **(1). Layering vs. Decomposition:**  Both methods utilize a similar stack of off-the-shelf vision foundation models (e.g., Grounding DINO, SAM, and Depth Pro) to parse the input image.
>
> The goal of our layering stage is to lift objects into **editable 3D proxies** that facilitate precise, 3D-grounded control through Blender. CAST uses these foundation models primarily to extract **conditioning signals** (partial point clouds). These signals are then fed into their "Occlusion-aware Object Generation" module to strictly automate the generation of watertight 3D meshes for every object.
>
> **(2). Generative Compositing vs. 3D Mesh Generation:** The most fundamental difference lies in the domain and timing of the generative process.
>
> Our diffusion model functions as a **refiner and neural renderer** at the **end** of the whole pipeline. It operates in pixel space, taking the rendered "target stream" and synthesizing a photorealistic result. This allows us to produce high-fidelity visuals even if the intermediate 3D proxies are coarse or imperfect. CAST’s generative model operates in **3D geometry space** as the **middle** step of the pipeline. It explicitly generates Signed Distance Fields (SDFs) and meshes using a 3D diffusion model, focusing on hallucinating missing geometry to create a static 3D asset.
>
> Note that while our default layering pipeline produces 2.5D meshes for efficiency, we also include an advanced option similar to CAST’s 3D mesh generation: using powerful **image-to-3D models** to generate high-quality, complete 3D meshes. This allows us to handle complex geometric transformations when necessary (e.g., as shown in Figure 8 of our paper). When generating complete 3D object meshes, we make use of the object’s 3D bounding boxes and the 2.5D partial reconstructions to determine their initial positions, while CAST proposes a sophisticated optimization-based algorithm to improve the physical realism/accuracy of the entire scene’s layout.
>
> **(3). Preferability:**  The choice between the two methods depends on the desired output.
>
> Our method excels when the goal is to produce **coherent, high-fidelity new visuals** that require precise, 3D-aware control. By decoupling control (Blender) from generation (Compositor), we enable flexible edits without needing perfect physics-ready meshes for every scene element.
>
> CAST is preferable when the goal is to export **high-quality 3D assets** for downstream applications such as physics simulations or robotics. CAST focuses on automatically recovering physically plausible, collision-free geometry via optimization constraints, which is critical for simulation but less essential for visual editing tasks.
>
> We will add discussions of CAST into the related works section of the paper.
>
> ----
>
> ### **2. More complicated examples**
>
> We provide additional results in **Figure 15 of the updated paper PDF**, including examples from the MOVI-E dataset with more than 10 objects. There are substantial object pose changes and camera viewpoint changes in those examples, and our method shows clear advantages in geometric correctness and appearance preservation.

---

### Official Review · Reviewer_14r8 · 2025-10-29

**Soundness:** 3
**Presentation:** 2
**Contribution:** 2
**Rating:** 4
**Confidence:** 3

**Summary:**

This paper introduces BlenderFusion, a novel framework for 3D-grounded visual editing and generative compositing. The core idea is to integrate the precise, 3D-aware control of a graphics engine like Blender with the powerful synthesis capabilities of diffusion models.

**Strengths:**

1. Clear Motivation and Strong Problem Formulation: The paper clearly identifies a significant and practical limitation in current generative AI: the lack of precise, 3D-aware control for complex, multi-object scene compositing. It effectively positions its contribution against existing methods (Table 1), clearly highlighting the gap it aims to fill.

2. Novel and Elegant Framework Design: The primary strength of this work lies in its core idea of decoupling 3D control from generative synthesis. By leveraging a mature graphics engine (Blender) for precise geometric manipulation and a diffusion model for photorealistic rendering, the framework provides an intelligent and practical solution.

3. Effective Training Strategies: The two proposed training strategies, "Source Masking" and "Simulated Object Jittering," are simple, well-motivated, and clever solutions to concrete problems. They effectively address the challenges of large-scale edits (e.g., object removal) and the entangled nature of object/camera motion in video datasets, which is critical for achieving robust, disentangled control.

**Weaknesses:**

1. Insufficient Detail on the Core Technical Novelty (Sec. 3.2): The paper's primary methodological contribution, the "Dual-stream Diffusion Compositor" in Section 3.2, is not described with sufficient clarity. The architecture is presented as a high-level black box, and the paper fails to provide a detailed diagram or explanation of the crucial "cross-stream interaction" mechanism. It is strongly recommended that the authors add a dedicated figure and more detailed text to fully articulate this component.

2. Engineering-Heavy Framework with Brittleness (Sec. 3.1): While Section 3.1 serves as an adequate description of the overall framework, it is fundamentally an engineering pipeline that chains together multiple existing, off-the-shelf models (e.g., Grounding DINO, SAM2, Depth Pro). This practical approach introduces significant brittleness, as a failure in any upstream component can compromise the entire process. Furthermore, the lack of novel components in this section concentrates the paper's entire scientific contribution into the insufficiently explained compositor, weakening the overall methodological robustness. A discussion on these limitations should be included.

3. Inconsistent Object Appearance Preservation: The qualitative results, while strong, reveal issues with preserving fine-grained object appearance and texture. For example, in Figure 5, the details on the yellow car change, and in Figure 10, the "Frosted Flakes" cereal box texture is visibly altered. This is likely an inherent limitation of "re-synthesizing" the object rather than copying pixels, perhaps due to information loss in the VAE or imprecise 2.5D geometry. The authors should acknowledge this limitation in their analysis and discuss its potential causes.

4. Missing Runtime and Scalability Analysis: The paper lacks any discussion of the framework's computational cost. The full pipeline involves multiple large models plus Blender rendering, suggesting it may be too slow for interactive use. Furthermore, the quality of the single-view 2.5D reconstruction is a critical bottleneck for complex or self-occluded objects. The authors should provide a runtime analysis (e.g., time per edit) in the appendix and discuss the scalability and quality trade-offs of their approach, including the mentioned option of using more costly image-to-3D models.

**Questions:**

Same as weakness.

---

> ### Author Response · Authors · 2025-11-24
> **Response to Reviewer 14r8**
>
> We thank the reviewer for the comments. We will address the concerns and questions one by one.
>
> ---
> ### **1. More details for the Dual-stream Diffusion Compositor**
>
> We have added a detailed illustration of the dual-stream diffusion compositor in **Fig. 13** and **Section F.1** of the revised PDF. We hope this clarifies the compositor model's architectural design.
>
> ---
>
> ### **2. Brittleness of the framework**
>
> We emphasize that our framework represents a novel meta-architecture designed to bridge traditional 3D engine workflows with the generative rendering capabilities of diffusion models. Rather than being a brittle and rigid pipeline, this modular integration is intentional, designed to ensure both synergy and extensibility.
>
> Regarding concerns about error propagation in a multi-stage design, our analysis indicates that the strong generative prior of the diffusion compositor effectively mitigates potential noise or imperfections from upstream components. This is confirmed by our extensive empirical results demonstrating robustness across real-world datasets (Objectron and Waymo), complex scenes with over 10 objects (MoVI-E, see more examples in **Fig.15** of the updated paper PDF), and cross-domain tasks (Fig. 1, 6, 8, and our demo page in the supplementary material), where our new approach significantly outperforms existing paradigms.
>
> Furthermore, we view this modularity as a strategic design feature. It allows our framework to evolve alongside rapid advancements in vision foundation models (e.g., the recent SAM3 or Depth Anything 3) and base diffusion models.
>
> We will incorporate the discussions into the paper.
>
> ----
>
> ### **3. Object appearance preservation**
>
> First, we note that the “Frosted Flakes’’ example appears in Fig. 5 rather than Fig. 10. Our result (last column) preserves texture details more faithfully than the baselines.
>
> To further quantify appearance consistency, we compute DINO and CLIP similarities across Objectron and Waymo Open Dataset (WOD), with the disentangled control and fine-grained compositing setup. For each object, we measure the similarity between the feature extracted from its source bounding box and the corresponding bounding box in the composited image:
>
> | Method | Objectron (mean ± std) |  | WOD (mean ± std) |  |
> | --- | --- | --- | --- | --- |
> |  | DINO | CLIP | DINO | CLIP |
> | Neural Assets | 0.7482 ± 0.0853 | 0.7170 ± 0.0983 | 0.8404 ± 0.0571 | 0.7709 ± 0.0540 |
> | BlenderFusion | **0.8658 ± 0.0523** | **0.8104 ± 0.0717** | **0.8906 ± 0.0591** | **0.8223 ± 0.0498** |
>
> Our method consistently outperforms Neural Assets across both datasets and metrics, demonstrating significantly stronger appearance and identity preservation.
>
> We agree that some appearance drift remains. As the reviewer notes, this is partly due to information loss in the Stable Diffusion VAE and imperfections in the 2.5D geometry and Blender renders. More broadly, maintaining fine-grained object identity during generative editing is an open challenge for the community. We will add a discussion of these limitations and their causes in the revised paper.
>
> ----
>
> ### **4. Running time analysis**
>
> We report inference times on the Objectron dataset using the same setup described in Table 2 and Fig. 4. The batch size is 12, we use 50 DDPM denoising steps, and all experiments were conducted on a single A100 GPU. The runtime breakdown is as follows:
>
> - **3DIT:** 6.16 s
> - **Neural Assets:** 10.92 s
> - **BlenderFusion:**
>     - Layering (SAM2.1 + Depth Pro): 0.6 s
>     - Blender scripts for automatic manipulation of 12 scenes: 3.0 s
>     - Diffusion compositor: 9.48 s
>     - **Total:** 13.08 s
>
> The Blender editing stage is highly efficient because the object meshes are lightweight; the renderings are intentionally coarse to provide only an accurate geometric prior while deferring high-frequency detail generation to the diffusion compositor (please refer to the video demo and supplementary webpage). Although our runtime is slightly longer than the baselines, the substantial performance improvement justifies this trade-off. The diffusion compositor is trained to operate on inherently noisy 2.5D reconstructions and effectively refines coarse renders, thus able to handle complex or self-occluded objects whose reconstructions are broken or noisy (as shown by many demo examples in the webpage in our supplementary material).
>
> For advanced editing in Fig. 8 (bottom), we use high-quality single-view image-to-3D models to obtain more detailed meshes. These operations enable sophisticated texture and geometry edits that baselines cannot handle. High-quality reconstruction methods such as Hunyuan 3D, Rodin, or Meshy typically require **2–5 minutes per scene**, so we apply them only to these advanced cases. All main experiments rely on efficient 2.5D lifting with SAM2 and Depth Pro.
>
> We will include this runtime analysis in the appendix and add a discussion of scalability and quality trade-offs.

---

> > ### Comment · Reviewer_14r8 · 2025-11-27
> > **Comment to Authors**
> >
> > Thank you for your detailed response. The additional clarifications regarding the model architecture and the runtime analysis have addressed several of my technical concerns.
> >
> > However, I still have a reservation regarding the usability of the proposed framework. Since the entire workflow is built upon Blender, it inherently imposes a significant barrier to entry, requiring users to possess a certain level of expertise in 3D graphics software. This steep learning curve may limit the framework's accessibility compared to other generative editing tools.
> >
> > That said, I acknowledge the value of the precise control this integration offers. I am currently inclined to raise my score, but I will wait to hear the perspectives of the other reviewers during the discussion period before finalizing my rating.

---

> ### Author Response · Authors · 2025-11-27
> **Response to Reviewer 14r8 (2)**
>
> We appreciate your follow-up and are glad the technical clarifications were helpful.
>
> Regarding usability, we would like to clarify that all operations used in our main experiments—object translation, rotation, rescaling, insertion/removal, and camera motion (Fig. 4, Fig. 5)—are performed entirely through **automatic Blender Python scripts**. These require no manual interaction and can run in exactly the same UI protocol (e.g., let the user input the control parameters) as prior 3D-aware editing methods such as 3DIT, Neural Assets, or training-free approaches like Diffusion Handles (Fig. 14 in the updated PDF). Blender’s UI is only used for highly customized or fine-grained edits, such as the advanced control and re-compositing cases in Fig. 6 and Fig. 8.
>
> Regarding the learning curve, we agree that Blender requires more expertise than standard, feed-forward generative tools. However, as a research effort, our primary goal is to revisit and demonstrate the validity of integrating traditional 3D engine controls with modern diffusion-based rendering.
>
> We view this work as a foundational step to empirically confirm this workflow. With this 'signal' established, we believe accessibility is a solvable engineering challenge; future efforts can develop high-level APIs or interactive plugins to abstract away the manual complexity, effectively lowering the barrier while retaining the precise control of the underlying 3D engine -- **this is actually happening now**, powered by the strong coding and general multimodal understanding capability of LLMs (e.g., the [BlenderMCP](https://github.com/ahujasid/blender-mcp) project).
>
> We hope this helps address your concern, and we appreciate your willingness to reconsider your score.

---

### Official Review · Reviewer_1PYq · 2025-11-02

**Soundness:** 3
**Presentation:** 3
**Contribution:** 1
**Rating:** 2
**Confidence:** 4

**Summary:**

The paper proposes a Blender-guided scene editing pipeline: segment and lift objects; perform multi-object edits, layout changes, and camera moves in 3D; then use a diffusion model to blend and harmonize the edited render with the original photo.

**Strengths:**

Clear, production-like workflow; easy to implement.

Results suggest better local control on some inserts.

**Weaknesses:**

**Key Baselines Omitted:**

- ZeroComp[1]: composites intrinsic layers (depth/normal/albedo/shading) and lets diffusion render the final image. Similar goal but without using Blender directly. But they use a rendering engine to give approx 3D compositing.
- DiffusionRenderer [2]: turns G-buffers into photoreal images/videos; direct alternative to “Blender render to diffusion fix.”
- 2D diffusion compositors: ObjectStitch [3], Thinking Outside the BBox [4], ControlCom [5], IMPRINT [6]: Generative Object Compositing by Learning Identity-Preserving already handle harmonization, identity preservation, and shadows/reflections without 3D renders. I just added a few of them, and there are plenty more papers on this theme.
- Multi-object, layout-controlled compositing: Multitwine [7] --  multi-object edits with text + layout control and interaction handling (occlusions, relations). This directly tests beyond single-object.
- 3D Copy-Paste [8]: physically plausible insertion (placement/collision/lighting) for real scenes; overlaps the paper’s claims on scene consistency.

Each of these baselines addresses challenges such as harmonization, identity preservation, and scene consistency using distinct strategies often without requiring full 3D modeling. Benchmarking against or discussing these alternatives in more depth would clarify how the proposed method advances or complements the current state of the art.

**Evaluation gaps for compositing**

The paper does not quantitatively evaluate several aspects: (1) background fidelity outside the edit mask, (2) identity preservation for inserted objects, (3) realism of contact shadows and reflections, and (4) multi-view consistency under small camera perturbations. These can be measured in controlled settings, e.g., using synthetic data from Blender.

**Robustness not tested:** The approach is not evaluated for common failure modes for this system's type of pipelines when relying on several components to get the final rendering. These include noisy masks, moderate pose or scale inaccuracies, challenging scene cases (e.g., clutter, high reflectance/specularities), and the domain gap from synthetic to real data.

**Ablations and cost:** It is unclear whether improvements are primarily from the 3D editing stage or the generative compositor. There is also an absence of runtime analysis -- how long does it take for the edit to be complete, etc, or failure case breakdown, leaves the practical utility and scalability unsure.

**References:**

[1] Zhang et al., ZeroComp: Zero-Shot Object Compositing from Image Intrinsics via Diffusion, WACV 2025.

[2] Liang et al., DiffusionRenderer: Neural Inverse and Forward Rendering with Video Diffusion Models, CVPR 2025.

[3] Song et al., ObjectStitch: Object Compositing with Diffusion Model, CVPR 2023.

[4] Canet Tarres et al., Thinking Outside the BBox: Unconstrained Generative Object Compositing, ECCV 2024.

[5] Zhang et al., ControlCom: Controllable Image Composition using Diffusion Model, arXiv 2308.10040.

[6] Song et al., IMPRINT: Generative Object Compositing by Learning Identity-Preserving, CVPR 2024.

[7] Canet Tarres et al., Multitwine: Multi-Object Compositing with Text and Layout Control, CVPR 2025.

[8] Ge et al., 3D Copy-Paste: Physically Plausible Object Insertion for Monocular 3D Detection, NeurIPS 2023.

**Questions:**

- Can you report SSIM/LPIPS outside the edit mask for all methods (yours and baselines)?
- Can you report DINO/CLIP similarity (inserted asset vs final composite) and show failure cases?
- How do you quantify contact-shadow and reflection realism? Will you provide a metric (e.g., overlap with a rendered reference) or a small user study, and add complex objects like glossy/metal scenes?
- Can you add: (a) Blender-only (no diffusion), (b) diffusion-only (no Blender edits), (c) replace Blender RGB with intrinsic layers (depth/normal/albedo/shading) to isolate what the compositor needs, (d) remove object-jitter and masking?
- Please provide a wall-clock breakdown (segmentation/lifting, rendering, diffusion), GPU hours for training/inference, memory, and throughput vs baselines.

---

> ### Author Response · Authors · 2025-11-24
> **Response to Reviewer 1PYq**
>
> We thank the reviewer for the effort in reviewing our paper. We address the questions and concerns below.
>
> ----
> ### **1. More Baselines**
>
> Note that Multitwine, ObjectStitch, ControlCom, and IMPRINT are **not** omitted; they are already discussed in our related works section.
>
> We will incorporate the remaining cited papers and clarify their connections. However, these methods focus on 2D or layout-based compositing and harmonization. While broadly related to visual editing, they are not directly comparable to our multi-object, 3D-aware control setting. For this setting, **3DIT** and **Neural Assets** remain the most relevant baselines, as they operate at the object level with explicit 3D transformations.
>
> We also include results for **Diffusion Handles** in **Fig. 14 and Sec. F.2 of the updated PDF**. Diffusion Handles is a training-free 3D-aware editing approach based on depth warping. As with several works mentioned by the reviewer, it is conceptually related but not designed for multi-object control or camera manipulation. As demonstrated by the results, such methods can fail badly under our experimental conditions and therefore do not serve as appropriate baselines for our evaluation.
>
> ---
>
> ### **2. Background PSNR/SSIM**
>
> We report background fidelity (pixels outside the foreground bounding boxes). BlenderFusion achieves clear gains over both baselines:
>
> |  | MoVI-E |  | Objectron |  | Waymo |  |
> | --- | --- | --- | --- | --- | --- | --- |
> |  | PSNR | SSIM | PSNR | SSIM | PSNR | SSIM |
> | 3DIT | 25.4176 ± 5.5152 | 0.8173 ± 0.1001 | 18.1703 ± 4.1974 | 0.6227 ± 0.1524 | 24.2250 ± 4.2690 | 0.6878 ± 0.1077 |
> | Neural Assets | 24.4946 ± 4.8583 | 0.7538 ± 0.1087 | 17.5029 ± 3.7932 | 0.5921 ± 0.1574 | 22.4614 ± 3.6833 | 0.6096 ± 0.1023 |
> | BlenderFusion | **29.7779 ± 6.0720** | **0.8772 ± 0.0744** | **19.7769 ± 4.8057** | **0.6875 ± 0.1459** | **25.1874 ± 4.5012** | **0.7229 ± 0.1024** |
>
> ---
>
> ### **3. DINO/CLIP Similarity (Inserted Asset vs Composite)**
>
> Table 2 already reports DINO similarity under the video-frame setting. We additionally compute DINO and CLIP similarity for the disentangled control and compositing tasks (Fig. 5/6 style). For each object, we compare features from the source bounding box to the corresponding region in the composited image:
>
> | Method | Objectron (mean ± std) |  | WOD (mean ± std) |  |
> | --- | --- | --- | --- | --- |
> |  | DINO similarity | CLIP similarity | DINO similarity | CLIP similarity |
> | Neural Assets | 0.7482 ± 0.0853 | 0.7170 ± 0.0983 | 0.8404 ± 0.0571 | 0.7709 ± 0.0540 |
> | BlenderFusion | **0.8658 ± 0.0523** | **0.8104 ± 0.0717** | **0.8906 ± 0.0591** | **0.8223 ± 0.0498** |
> |  |  |  |  |  |
>
> The results confirm that BlenderFusion consistently outperforms Neural Assets in terms of object appearance/semantics preservation.
>
> ----
>
> ### **4. Contact Shadows and Reflections; Complex objects like glossy/metal scenes**
>
> Modeling physically accurate shadows/reflections or complex glossy/metal scenes is outside the scope of this paper and is addressed by neither our method nor the baselines.
>
> ---
>
> ### **5. Ablation Studies**
>
> We respectfully note that the requested ablations are already addressed by the results in our existing experiments (Fig. 7 and Table 3). Concretely,
>
> - **(a) Blender-only (no diffusion):** Blender renders are intentionally coarse (Fig. 2 and supplementary demos) and do not produce photorealistic outputs; consequently, this setting does not isolate a meaningful component of our method.
> - **(b) diffusion-only (no Blender edits):** This is equivalent to the “dual-stream w/ Blender” ablation (Fig. 7, col. 4).
> - **(c) intrinsic layers instead of Blender RGB:** This is captured by the depth+segmentation input ablation (Fig. 7, col. 3).
> - **(d) remove object jitter or masking**: These are evaluated in Fig. 7, columns 5 and 6."
>
> ---
>
> ### **6. Runtime, Training Cost, and Memory**
>
> **Training cost.** Under the training setup detailed in Sec. 4.1 and Sec.B.1, the training time and GPU costs are:
>
> - **3DIT:** ~30 hours, ~40GB per GPU
> - **Neural Assets:** ~40 hours,  ~60GB per GPU
> - **BlenderFusion (compositor only):** ~48 hours, ~ 65GB per GPU
>
> **Inference.** The results below are measured on Objectron with batch size 12, 50 DDPM sampling steps, and a single A100 GPU:
>
> - **3DIT:** 6.16 s
> - **Neural Assets:** 10.92 s
> - **BlenderFusion:**
>     - Layering (SAM2.1 + Depth Pro): 0.6 s
>     - Blender scripting (12 scenes): 3.0 s
>     - Diffusion compositor: 9.48 s
>     - **Total:** 13.08 s
>
> All approaches fit within 24GB of memory with BF16 inference.

---

> > ### Comment · Reviewer_1PYq · 2025-11-26
> > **Response to author's rebuttal**
> >
> > Thanks to the authors for taking the time to address the concerns in their rebuttal. Thanks for adding clarifications on background change and other ablations and DINO/CLIP similarity scores.
> >
> > My biggest concern was the **Key Baselines Omitted**. Thank you for adding the Diffusion Handles baseline (I named a few of them for *comparison*, and Diffusion Handles is reasonable), which partly addresses the diffusion-based generative compositing. However, it still remains unclear why ZeroComp and Diffusion Renderer are not compared. Both of these are the most relevant and strongest baselines, especially ZeroComp, as it uses the concept of using a rendering engine as guidance for compositing (see their Fig 2 and see their section 3.2 "For the objects we intend to insert (Fig. 2, middle), we use the Blender [16] graphics engine to render its intrinsic layers i_obj, except shading which is unknown."), and importantly, their **training does not require any paired images (with and without composites; a harder task)**. They leverage learned intrinsic maps and rendering engine guidance to position objects in a zero-shot manner at test time, correct their shading, and partially refine the surrounding scene using the expanded mask. This insertion can be easily manipulated in Blender. For example, objects can be rotated, and multiple objects can be added through the rendering engine if necessary, with corresponding intrinsic maps further extracted. Similarly, DiffusionRenderer is also capable of compositing. These models are 3D-aware in that they predict intrinsic components related to the scene’s 3D aspects (geometry, appearance, and lighting). However, without a comparison of when and why these methods might fail, or evidence that BlenderFusion outperforms them, the contribution of the overall paper appears incremental.

---

> ### Author Response · Authors · 2025-11-26
> **Response to Reviewer 1PYq -- Why ZeroComp and DiffusionRenderer are not directly comparable**
>
> Thank you for the follow-up discussion. We appreciate the chance to clarify why **ZeroComp** and **DiffusionRenderer** operate under task settings fundamentally different from ours, making them not meaningful baselines for our evaluation.
>
> ---
>
> ### 1. Our task formulation (Table 2 / Fig. 4)
>
> **Inputs:**
>
> - A single real image containing **multiple objects**
> - For each object: its current 3D pose and a target 3D pose
> - A relative camera motion
> - **No 3D mesh** is provided for any object
>
> **Output:** A photorealistic image where **the existing objects in the input** follow the specified 3D transformations.
>
> This is the task that **3DIT** and **Neural Assets** are designed for. In extended settings (Figs. 1, 6, 8), we also support multi-image re-compositing and intra-object edits that 3DIT and Neural Assets *cannot do*, still without assuming ground-truth meshes.
>
> ---
>
> ### 2. Why ZeroComp is not applicable
>
> ZeroComp operates under a fundamentally different input–output setup:
>
> - **Inputs:** a background image and an **external 3D object mesh**
> - **Output:** a composite image blending the background and the mesh, guided by intrinsic layers rendered via Blender
>
> ZeroComp never edits or manipulates the objects already present in an input image. Its training “**does not require any paired images”** because its task setting **does not** require recovering object geometry from images (a clean 3D mesh is given as input) or controlling the camera viewpoint.
>
> More specifically, it cannot be applied meaningfully to our setting because:
>
> - It cannot edit or re-compose **objects inside the image**
> - You might say that they can use the lifted 2.5D meshes from our layering stage, but their method cannot operate on *noisy 2.5D geometry* estimated from monocular depth
> - It does not support **camera viewpoint changes**
> - Most importantly, it **assumes explicit clean 3D meshes**, which is not available in our problem setup
>
> Since our evaluation requires transforming the objects *derived from the image itself*, we do not see a principled way to adapt ZeroComp to our setting. If the reviewer has a concrete proposal for doing so, we are happy to evaluate it.
>
> ---
>
> ### 3. Why DiffusionRenderer is also not applicable
>
> DiffusionRenderer similarly assumes known 3D object meshes and performs inverse/forward rendering of intrinsic components for relighting and insertion. It does not edit or re-render the objects inside the input image and does not operate in our multi-object, pose-controlled, camera-moving setup.
>
> ---
>
> ### 4. Summary
>
> ZeroComp and DiffusionRenderer are both 3D-aware compositing methods, but:
>
> - They assume **explicit and clean 3D meshes** as inputs, and focus on inserting **external** objects coherently into a background image
> - They do not support editing or re-composing **the objects inside the input image**
> - They do not operate in **multi-object, pose-controlled, camera-changing** scenarios
>
> Our task requires controlling and re-rendering **existing objects** from one or multiple images, without access to ground-truth meshes—an entirely different setting.
>
> If the reviewer can provide a specific way to adapt these pipelines to manipulate the objects already present in an input image, we are happy to run the comparison.

---

> > ### Comment · Reviewer_1PYq · 2025-11-28
> > **Re ZeroComp and Diffusion Renderer**
> >
> > Disagree with the authors that they need clean meshes for these baselines. L197-198 in their own paper says that "we optionally use image-to-3D models (Xiang et al., 2024; Zhao et al., 2025) to produce complete object meshes." If the authors have already used image-to-3D models in their work, then it should be straightforward to construct the above-mentioned baseline. Specifically, one can readily lift objects to 3D using these models, manipulate them in Blender, and subsequently run the result through ZeroComp or Diffusion Renderer for cleanup. In addition, Diffix3D (Wu et al., CVPR 2025) can be applied for post-processing to remove any remaining artifacts—though this step might not always be necessary, as strong diffusion priors tend to generate realistic images by default. These combined methods remain both strong and valid baselines and do not require any new training. Moreover, the core contribution of using Blender + Diffusion has already been established. In ZeroComp, for example, Blender performs the heavy lifting for object placement, after which the Diffusion Model refines the rendered image.

---

> ### Author Response · Authors · 2025-12-01
> **Response to Reviewer 1PYq -- ZeroComp is structurally/functionally invalid as a baseline**
>
> Thank you for the follow-up. We respectfully disagree that ZeroComp or DiffusionRenderer are directly applicable baselines. While these methods share components (Blender and diffusion models), this **does not** imply comparable task settings or capabilities. Our experimental findings further support this distinction.
>
> ### 1. More clarifications:  why ZeroComp / DiffusionRenderer does not match our setting
>
> Neither method is designed to modify objects already present in an input image. Their formulation assumes **clean, complete 3D meshes** as inputs and focuses on inserting them into a background image. Applying them to our setting introduces several non-trivial challenges:
>
> - **Geometric alignment with the input image.**    Our task requires aligning reconstructed object meshes to the precise 3D pose in the input image. We rely on 3D bounding boxes and ICP refinement—functionality not present in ZeroComp or DiffusionRenderer.
>
> - **Handling existing objects in the scene.**    These methods assume objects come exclusively as external 3D assets. They do not handle modifying or removing objects already present in the image, which is central to our problem.
>
> - **Robustness to noisy 3D reconstruction.**    Even strong image-to-3D models produce imperfect meshes, especially in cluttered or multi-object scenes (e.g., Waymo). Our compositor is trained to correct these imperfections. ZeroComp, trained solely on synthetic data with clean FG–BG separation, is not designed for such cases.
>
> ### 2. Despite the mismatch, we respect the reviewer's comment and ran the experiments
>
> Following the reviewer’s suggestion, we adapted ZeroComp for our task settings; all the following adaptations are **not part of the original ZeroComp pipeline**:
>
> - Using our layering stage for lifting and 3D alignment, including running an image-to-3D model for all the objects, and doing proper alignment to the original image.
> - Inpainting the original image to remove existing objects.
> - Extending ZeroComp to handle multiple objects, as the public implementation only supports single-object insertion.
>
> Evaluation is limited to **disentangled object control**, since ZeroComp does not support camera motion.
>
> ### 3. Results
>
> Please see Figure 16 in the updated PDF for the results:
>
> - **Objectron:**  ZeroComp yields reasonable but clearly inferior results compared to Neural Assets and BlenderFusion. Lighting and shading mismatches remain prominent due to the synthetic training domain. As discussed above, training on real-world images is non-trivial using the existing ZeroComp pipeline, since it assumes clean separation of foreground objects and background image, which does not hold for real-world data.
>
> - **Waymo Open Dataset:**   Performance drops sharply. With many objects and noisy image-to-3D reconstructions, ZeroComp fails to maintain consistent geometry or appearance.
>
> ### 4. Summary
>
> ZeroComp (or DiffusionRenderer) is conceptually related in terms of the use of Blender + diffusion, but they address a different problem—**inserting external 3D assets and keeping coherent lighting and shading**, not editing or recomposing objects directly from real images. They lack support for multi-object edits, camera motion, and robustness to noisy 3D lifting, all of which are central to our setting. Even with substantial adaptation effort, ZeroComp's performance remains far below our current baseline, Neural Assets.

---

### Author Response · Authors · 2025-12-01
**A summary of the rebuttal**

We sincerely thank the reviewers, AC, and SAC for their work during this unusual review cycle. The purpose of this message is to (1) clarify key technical misunderstandings that may undervalue our contribution, and (2) provide a concise summary of the rebuttal updates and reviewer interactions.

---

### 1. Key technical misunderstandings

A recurring assumption across R1 and R4 is that methods that share generic components (Blender, diffusion models, or object-centric processing) imply similar contributions and task scope. This is not the case. These misunderstandings could lead to misjudging our technical contributions and the relevance of certain baselines.

**(R1-1PYq) “Missing baselines” (ZeroComp, DiffusionRenderer)**:

R1 assumes that ZeroComp/DiffusionRenderer are directly comparable baselines simply because they use Blender + diffusion. This overlooks the fundamental task mismatch:
- These methods ASSUME **clean, complete 3D meshes** as input. As demonstrated in our experiments, noisy meshes cause the entire pipeline to fail.
- They focus on **object insertion**, NOT editing objects *already in an image*. While insertion can be achieved using strong diffusion generative priors, editing requires both the 3D geometry of existing objects and a prior.
- They do NOT support **multi-object control**, **camera motion**, or robustness to **noisy 2.5D lifting**.

To verify this empirically, we **adapted and evaluated ZeroComp** under the disentangled-control setting (Fig. 16). However, even with considerable engineering efforts for adaptation, the performance remained far below the BlenderFusion, clearly demonstrating that these methods are **not meaningful baselines** for our task.

**(R4-k2sd) “Limited novelty and scalability”**:

R4 interprets our method as either similar to previous object-centric modelling works (e.g., Neural Asset) or a simple variant of tool-use (e.g., ScenCraft) pipelines. Neither interpretation is accurate, as we have not positioned our proposal within either perspective.

Instead, our proposal acts as a practical bridge between **traditional 3D workflows** and **modern generative models**. We leverage Blender for accurate scene layout and geometric control, while relying on diffusion models for high-quality rendering. The core technical contribution enabling this is our **dual-stream diffusion compositor**, which introduces novel architecture change (Sec 3.1) and training strategies (Sec 3.2)  to robustly handle inherently noisy Blender proxies. This approach unlocks distinct advantages over previous SOTA baselines (3DIT, Neural Assets) including 1) multi-object control and multi-image recomposition, 2) camera-motion edits, and 3) advanced part-level/object-internal edits (e.g., Fig.1 and Fig.8).

In terms of scalability and robustness, we have extensively validated our performance on three challenging datasets spanning diverse domains (Objectron, WOD, and MOVI-E). Crucially, our modular meta-architecture is inherently future-proof; it allows for the seamless integration of stronger vision foundation models (e.g., SAM3, Depth Anything3) and upgraded base diffusion models without requiring pipeline modifications.

---

### 2. Summary of rebuttal updates

For clarity, we summarize the new experiments and analyses added during the rebuttal (all in **Sec. F** of the updated PDF for easier and stable cross-reference):

- **(R4-k2sd, R1-1PYq)** Added **Diffusion Handles** (Fig. 14) and **ZeroComp** (Fig. 16) baselines. Results confirm these methods are not competitive or appropriate baselines.
- **(R2-14r8)** Added a detailed architectural diagram and explanation of the dual-stream diffusion compositor (Fig. 13).
- **(R1, R4)** Added **background-only PSNR/SSIM** evaluation (Table 4).
- **(R1, R2)** Added **foreground-only DINO and CLIP similarity** evaluation for appearance preservation (Table 5).
- **(R1, R2, R4)** Added **training-time, inference-time, and memory** breakdowns (Sec. F.6).
- **(R3-jWo3)** Added additional results on complicated MoVI-E scenes with 10+ objects (Fig. 15).

---

### 3. Brief summary of reviewer interactions

- **R1-1PYq:** Remaining concern centered on “missing baselines.” We provided a concrete ZeroComp evaluation (Fig. 16), which supports our argument that ZeroComp/DiffusionRenderer are not meaningful baselines.
- **R2-14r8:** Stated that “several concerns are resolved.” The remaining concern is the usability and learning curve of Blender. We clarified that all basic edits are automated, and manual Blender use is required only for highly customized cases, which can also be automated in the future with LLM agents and engineering efforts.
- **R3-jWo3, R4-k2sd:** No follow-up comments before the portal closed.

---

We hope this message helps clarify the main technical issues raised during the review process and how we addressed them. Thank you again for your time and effort during this challenging review cycle.

---

### Meta-Review · Area_Chair_3A8R · 2026-01-07

**Summary:**

This paper proposed a three-step framework for 3D-grounded visual editing and generative compositing that integrates 3D graphics software and 3D representation in the workflow. The major concern of the reviewer are: 1) the novelty of the method; 2) how scalable the method is; 3) missing baselines.

**Reviewer Concerns:**

The authors have partially addressed these concerns, and some reviewer is inclined to raise the score. However, several issues still remain:
1) The paper stitches a number of tools (segmentation, image-to-3D, blender, etc.), and it looks like the major scientific novelty lies on the diffusion-based compositor.
2) It's unlikely the method can be scaled up due to the complexity of the pipeline.

**Reviewer Scores:**

Although one reviewer proposed to raise the score, I don't think there will be a score raising after the reviewer discussion period.

---

### Decision · Program_Chairs · 2026-01-26

Reject